# The endosomal Q-SNARE, Syntaxin 7, defines a rapidly replenishing synaptic vesicle recycling pool in hippocampal neurons

Yasunori Mori[1,3✉], Koh-ichiro Takenaka[1], Yugo Fukazawa [2] & Shigeo Takamori[1✉]

Upon the arrival of repetitive stimulation at the presynaptic terminals of neurons, replenishment of readily releasable synaptic vesicles (SVs) with vesicles in the recycling pool is important for sustained neurotransmitter release. Kinetics of replenishment and the available pool size define synaptic performance. However, whether all SVs in the recycling pool are recruited for release with equal probability and speed is unknown. Here, based on comprehensive optical imaging of various presynaptic endosomal SNARE proteins in cultured hippocampal neurons, all of which are implicated in organellar membrane fusion in non-neuronal cells, we show that part of the recycling pool bearing the endosomal Q-SNARE, syntaxin 7 (Stx7), is preferentially mobilized for release during high-frequency repetitive stimulation. Recruitment of the SV pool marked with an Stx7-reporter requires actin polymerization, as well as activation of the $Ca^{2+}$/calmodulin signaling pathway, reminiscent of rapidly replenishing SVs characterized previously in calyx of Held synapses. Furthermore, disruption of Stx7 function by overexpressing its N-terminal domain selectively abolished this pool. Thus, our data indicate that endosomal membrane fusion involving Stx7 forms rapidly replenishing vesicles essential for synaptic responses to high-frequency repetitive stimulation, and also highlight functional diversities of endosomal SNAREs in generating distinct exocytic vesicles in the presynaptic terminals.

[1] Laboratory of Neural Membrane Biology, Graduate School of Brain Science, Doshisha University, Kyoto, Japan. [2] Division of Brain Structure and Function, Research Center for Child Mental Development, Life Science Innovation Center, School of Medical Science, University of Fukui, Fukui, Japan. [3] Present address: Department of Biochemistry, Faculty of Medicine, University of Yamanashi, Yamanashi, Japan. ✉email: yasmori@yamanashi.ac.jp; stakamor@mail.doshisha.ac.jp

Neurons translate electrical stimuli to chemical signals in the form of neurotransmitters, which are transmitted to target cells by the exocytic fusion of neurotransmitter-filled vesicles termed synaptic vesicles (SVs). Hundreds to tens of thousands of morphologically homogeneous neurotransmitter-filled SVs are clustered at presynaptic terminals, potentially capable of releasing their contents upon the arrival of electrical stimuli[1]. Yet, only a fraction of all SVs is usually available for stimulus-dependent release[2,3] (but see ref. [4]). During the last few decades, the three-pool model has emerged, in which neurotransmitter-filled synaptic vesicles are classified into three functionally distinct pools: the readily releasable pool (RRP), the recycling pool that replenishes the RRP during sustained activity, and the reserve pool (RP, also referred to as the resting pool) that rarely participates in recycling[2,3]. Despite intensive research, however, the molecular underpinnings of the three pools have remained elusive.

Exocytic fusion of SVs to the plasma membrane is mediated by the formation of so-called SNAP-receptor (SNARE) complexes, consisting of vesicular SNARE synaptobrevin (Syb)/VAMP2 and two target SNAREs Syntaxin 1 (Stx1) and SNAP-25[5]. Considering the extraordinary speed of SV exocytosis in response to action potentials (typically within several ms), the RRP is believed to dock physically to the plasma membrane prior to fusion and the canonical neuronal SNAREs described above are responsible for their docked status[2,3]. Aside from the necessity of the dominant SNAREs for the final step of SV exocytosis, non-canonical SNAREs that mediate fusion reactions between intracellular organelles in non-neuronal cells are not only present but also enriched in SVs purified from rat brains[6] (Supplementary Fig. 1), albeit at lower expression levels than the canonical SNAREs[7], making them likely candidate protein constituents in functionally distinct subsets of SVs, such as the recycling and reserve pools. Indeed, in addition to synapsins, which are reportedly associated with RP[8], endosomal SNAREs are also present in distinct vesicle pools, namely in RP or spontaneously releasing vesicles (VAMP7 or vti1a), both of which are somewhat refractory to evoked release[9,10]. Similarly, several endosomal SNAREs including vti1a, Stx6, and Stx12/13, also engage in SV recycling and may supply RRP[11], and VAMP4 is directed to the asynchronous SV pool[12]. However, whether and/or to what extent these presynaptic endosomal SNAREs are present within the recycling pool and consequently confer unique properties upon it, has not been determined.

Aside from the three functionally distinct SV pools described above, accumulating evidence suggests that even within the same presynaptic terminals, distinct types of SVs coexist. For instance, vesicular glutamate transporters (VGLUTs) and vesicular monoamine transporter 2, which are responsible for uptake of glutamate and biogenic monoamines, respectively, into SVs, are present on different populations of SVs[13,14]. In addition to their distinct exocytosis kinetics at a fixed stimulation frequency, exocytotic kinetics of respective SVs during repetitive stimulation appeared to differ depending on stimulation frequencies[14], suggesting the existence of various SV recycling pools with distinct release probabilities. Although SV biogenesis mediated by adaptor protein (AP)-3 is a key step in the formation of functionally distinct SV pools[14], it remains unknown whether differences in their molecular compositions are due to minor SV proteins, such as presynaptic endosomal SNAREs.

To gain deeper insights into the contribution of endosomal SNAREs in SV recycling, we monitored behaviors of presynaptic endosomal SNAREs conjugated C-terminally with pHluorin (also referred to as SEP)[15,16] in a comprehensive manner in response to various stimulation protocols. We found that among endosomal SNAREs, Stx7 is exclusively sorted to a subpopulation of the SV recycling pool that responds preferentially to high-frequency repetitive stimulation. Rapid recruitment of this pool is activated by a $Ca^{2+}$/calmodulin-mediated pathway, and also depends on actin dynamics, resembling a previously reported SV population that rapidly replenishes RRP after synaptic depression[2]. Notably, blockade of Stx7 function by overexpressing its N-terminal domain selectively abolishes this pool. Thus, our results reveal a hitherto unrecognized mechanism by which the rapidly replenishing SV recycling pool that sustains synaptic transmission during high-frequency stimulation (HFS) is generated by endosomal fusion events involving Stx7 during SV recycling.

## Results

**Endosomal SNARE-SEPs localize in distinct types of presynaptic vesicle compartments.** To explore the recycling properties of SVs carrying presynaptic endosomal SNAREs under various stimulation protocols, we utilized a pHluorin reporter (super-ecliptic pHluorin: SEP) fused to the luminal C-terminal portions of individual SNARE proteins (Fig. 1a). These include five Syntaxin (Stx) family members (Stx6, 7, 8, 12/13, and 16), all of which were highly enriched in the pure SV fraction[6] (Supplementary Fig. 1). We also included two additional endosomal SNAREs, vti1a and VAMP7, which reportedly mediate spontaneous release rather than stimulus-dependent evoked release[9,10]. As a control for authentic SV residents, two widely used SEP constructs, SypHy (Synaptophysin (Syp) fused with SEP) (Fig. 1a) and VGLUT1-SEP were used[17,18]. These SEP reporters have been widely used to monitor SV recycling, owing to their pH dependence, by which SEP fluorescence is quenched in the acidic lumen of SVs, and de-quenched upon exocytosis by exposure to the neutral extracellular pH (Fig. 1b−d). When SypHy was lentivirally transduced into cultured hippocampal neurons[19,20], it was sorted to presynaptic compartments, evidenced by immunostaining with anti-Syb2 antibody (Fig. 1b). When neurons were electrically stimulated repetitively at 10 or 40 Hz (300 or 200 action potentials (APs)), SypHy fluorescence robustly increased in response to the stimuli, and >30−40% of all SypHy molecules, estimated by application of 50 mM $NH_4Cl$ at the end of the recordings, was engaged in exocytosis by the end of stimulation (Fig. 1d−f). On the other hand, although presynaptic endosomal SNARE-SEPs were similarly co-localized with Syb2 (Fig. 1e), the vast majority of endosomal SEPs showed relatively smaller responses compared to SypHy (Fig. 1e, f; <25%) and in some cases, responses upon 40 Hz stimulation were significantly smaller than those elicited at 10 Hz (e.g., Stx8-SEP, Stx16-SEP) (Fig. 1e, f). Strikingly though, Stx7-SEP exhibited unique behavior, i.e., stimulation at 10 Hz failed to elicit a clear fluorescence increase, while 40 Hz stimulation caused a robust increase up to ~40% (Fig. 1e, f). Quantitative comparisons of all SEP-reporters in response to 10 and 40 Hz stimulation (with 200 APs) revealed that Stx7-SEP exhibited an ~8-fold fluorescence increase at 40 Hz compared to 10 Hz (Fig. 1e, f). Overall, these results indicate that different presynaptic endosomal SNARE-SEPs localize to distinct membrane compartments that are capable of recycling in an activity-dependent manner, to different degrees, at presynaptic terminals.

**Only Stx7-SEP exclusively localizes to Syb2-positive SVs.** Given distinct recycling behaviors of presynaptic endosomal SNARE-SEPs compared to SypHy, we wondered to what extent these endosomal SNARE-SEPs are sorted into functionally fusion-competent SVs. To test this, SEP responses were monitored after pretreatment with tetanus toxin (TeNT), which enzymatically cleaves Syb2/VAMP2[21] (also cleaves VAMP1 or VAMP3/Cellubrevin)[22,23], thereby terminating SV exocytosis[24]. The robust response of Stx7-SEP at 40 Hz was completely blocked by TeNT treatment, as was observed for SypHy and VGLUT1-SEP, indicating that Stx7-SEP was sorted into Syb2/VAMP2-positive,

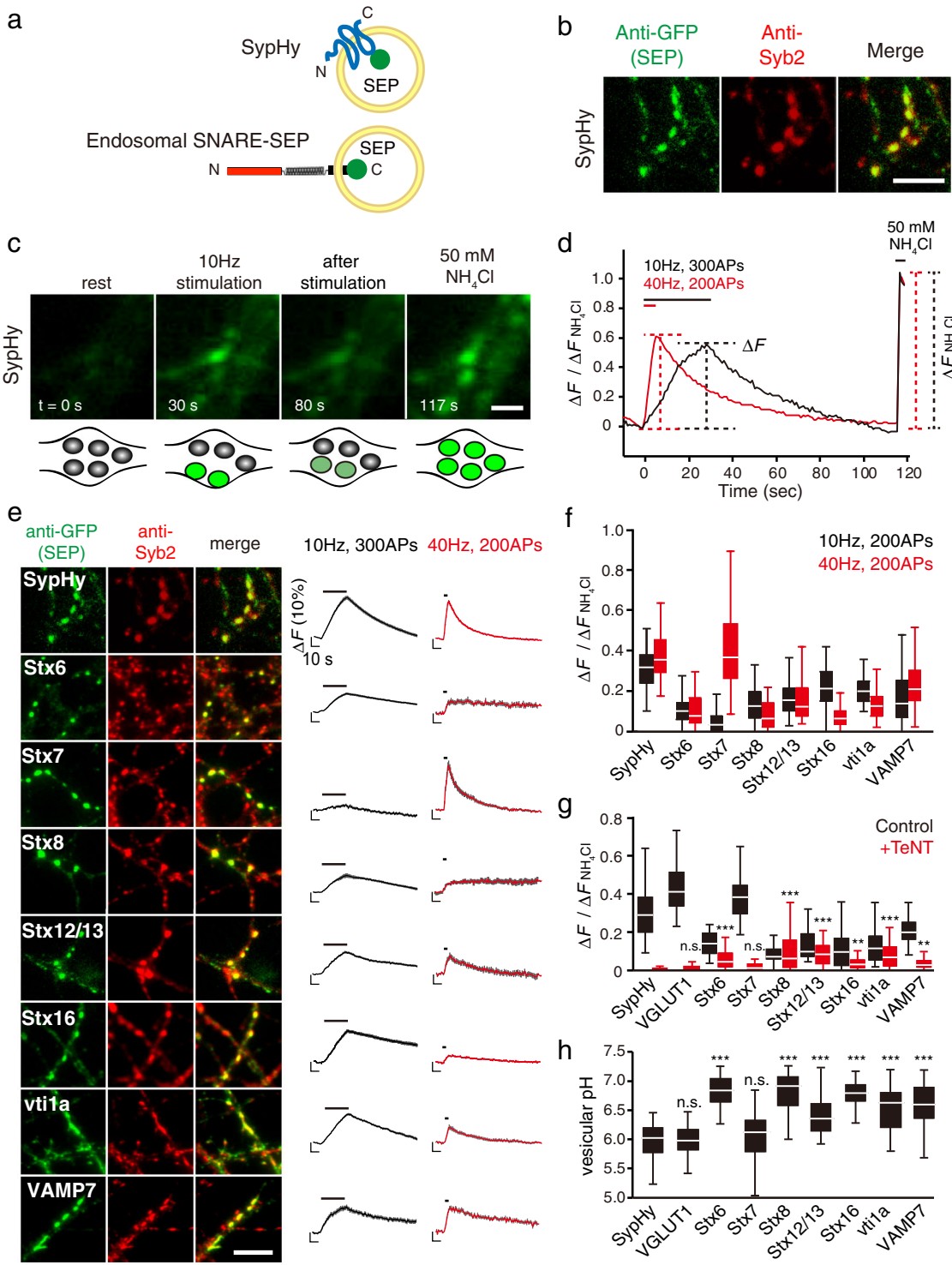

functionally competent SVs (Fig. 1g; Supplementary Fig. 2). By contrast, responses of other endosomal SNARE-SEPs were not completely abolished by TeNT treatment, but were only diminished to varying extents (Fig. 1g and Supplementary Fig. 2). We also examined luminal pH of intracellular compartments bearing SEP-reporters by sequential application of an acidic solution (pH 5.5) and a 50 mM $NH_4Cl$ solution[25], and subsequent electrical stimulation to ensure active synapses (Fig. 1h and Supplementary Fig. 3). These analyses revealed that the vesicular pH of Stx7-laden vesicles (mean ± standard error of the mean (s.e.m.), 6.09 ± 0.05) was not significantly different from those of SypHy (mean ± s.e.m., 5.96 ± 0.04) and VGLUT1-SEP (mean ± s.e.m., 5.98 ± 0.05), whereas those of other endosomal SNARE-SEP-laden vesicles were significantly higher (>6.40) (Fig. 1h). We also examined additional putative SNARE pairs of Stx7 suggested in non-neuronal cells, including VAMP8 and vti1b[26,27]. When VAMP8-SEP and vti1b-SEP were lentivirally transduced into neurons, they rarely co-localized with Syb2/VAMP2-positive puncta, and only minor portions of punctate structures labeled with these SNAREs responded to repetitive stimulation at 40 Hz, 200 APs (~11% of VAMP8-SEP-positive puncta and ~0.7% of vti1b-SEP-positive puncta showed fluorescence increases beyond

**Fig. 1 Comprehensive monitoring of presynaptic endosomal SNARE-SEPs reveals their characteristic recycling behavior. a** Cartoon representation of SypHy (Synaptophysin-SEP) and an endosomal-SNARE-SEP. **b-d** Strategies for comprehensive characterization of presynaptic SNARE-SEPs. After SEP constructs were lentivirally transduced in cultured hippocampal neurons, distributions of SEP-fused proteins and their responses to repetitive electrical stimulation were analyzed by immunocytochemistry (**b**) and fluorescence live imaging (**c**, **d**). Representative results obtained for SypHy are shown. The SEP was visualized with anti-GFP antibody, whereas locations of presynaptic boutons were identified with anti-synaptobrevin 2 (Syb2) antibody. To estimate fractional responses of each SEP construct, 50 mM $NH_4Cl$ (pH7.4) was applied at the end of recordings, which revealed the total expression of SEP-fused proteins at individual boutons. Scale bars indicate 5 μm in (**b**) and 2 μm in (**c**). **e** Synaptic localization and stimulus-dependent recycling of endosomal SNARE-SEPs upon 10 Hz or 40 Hz stimulation. Left images are representative images of each SEP-fused protein, co-stained with anti-Syb2 antibody. Scale bar indicates 5 μm. Right traces show average fluorescence of individual SEP-fused proteins upon 10 Hz (300 APs; black) and 40 Hz (200 APs; red) stimulation. Fluorescence was normalized to those during $NH_4Cl$ application. Data are averages of 50−200 boutons. **f** A box-whisker plot showing the peak fluorescence of the respective SEP probes at 200APs of 10 Hz (black) and 40 Hz stimulation (red). The boxes, the white lines in the boxes, and the whiskers in this plot and hereafter indicate the first and third quartiles, the medians, and the minimum and maximum values, respectively. **g** A box-whisker plot showing the effect of TeNT pretreatment on recycling of SEP probes in comparison to SypHy at 40 Hz stimulation (200 APs) (see also Supplementary Fig. 2). Data were obtained from 35−100 boutons. *P*-values indicate n.s $p > 0.05$, ** $p < 0.01$, and *** $p < 0.001$ in comparison to SypHy after TeNT treatment (Student's *t*-test). **h** A box-whisker plot showing luminal pHs of vesicle compartments carrying the respective SEP probes calculated from experiments in Supplementary Fig. 3. *** $p < 0.001$ indicates *p*-values in comparison to vesicular pH of SypHy (Student's *t*-test).

5% of total SEP fluorescence revealed by $NH_4Cl$ application, whereas ~77 and ~94% of Stx7-SEP-positive puncta and SypHy-positive puncta, respectively, showed responses beyond 5% of their total fluorescence) (Supplementary Fig. 4). Thus, these results collectively demonstrate that, among the presynaptic endosomal SNAREs examined here, Stx7 is peculiar in that it exclusively localizes to a subpopulation of the recycling pool containing genuine, fusion-competent SVs that respond preferentially to HFS.

**Exocytosis of Stx7-SEP vesicles requires HFS.** One of the most striking features revealed by our comprehensive SEP imaging analysis (Fig. 1e, f) was that Stx7-SEP responded preferentially to HFS at 40 Hz, but rarely responded to 10 Hz stimulation. To confirm this phenomenon and to avoid any bias possibly due to functional heterogeneity intrinsic to each bouton or culture preparation, we continuously monitored changes in SEP fluorescence at individual boutons with various stimulation frequencies ranging from 5 to 40 Hz at 5 min intervals (Fig. 2a). Unlike SypHy, which reliably exhibited a robust exocytotic fluorescence increase irrespective of stimulation frequency (Fig. 2a, top), Stx7-SEP hardly responded during 5 Hz or 10 Hz stimulation, while it showed robust fluorescence increases at 20 and 40 Hz in the same boutons (Fig. 2a, bottom). We ruled out the possibility that stimulus-dependent increases in Stx7-SEP fluorescence resulted from vesicle fusion to neutral intracellular compartments during HFS, since the application of an acidic solution (pH 5.5) right after cessation of stimulation largely quenched the fluorescence Stx7-SEP (Supplementary Fig. 5), as was observed in the case of SypHy[19].

We then wondered if overexpression of Stx7-SEP simply attenuated SV exocytosis in general, for instance by inactivating $Ca^{2+}$ channels, or if Stx7-SEP localized at non-synaptic areas where non-SV-type secretory vesicles that responded only to HFS were preferentially recorded. To explore these possibilities, we co-expressed Stx7-SEP and Syp-mOr, in which a pH-sensitive orange fluorescent protein, mOrange2, was fused to the luminal region of synaptophysin, instead of SEP[19,20], and restricted the analysis to Syp-mOr responding areas (Supplementary Fig. 6). Yet, Stx7-SEP rarely responded to 10 Hz stimulation, but exhibited a drastic increase in response to 40 Hz stimulation with this analysis (Supplementary Fig. 6).

We next wondered if more prolonged stimulation at low frequencies (5 and 10 Hz) could induce exocytosis of Stx7-SEP-laden vesicles. Since endocytosis and subsequent re-acidification of SVs during prolonged stimulation might mask the SEP fluorescence increase due to SV exocytosis[28], we blocked re-

acidification of endocytosed vesicles with 2 μM bafilomycin A1 (Baf), a potent inhibitor of vacuolar-type $H^+$-ATPases (V-ATPases) (Fig. 2b). We adopted 600 action potentials (APs) because that was sufficient to induce exocytosis of the entire recycling pool in cultured hippocampal neurons[29]. In the case of SypHy, fluorescence signals plateaued at the same levels (about 40−50%) irrespective of stimulation frequencies (5−40 Hz) (Fig. 2c). In stark contrast, Stx7-SEP exhibited similar exocytic properties as SypHy at 20 and 40 Hz, whereas it showed much lower and slower responses during 5 and 10 Hz stimulation (Fig. 2c). Comparisons between SypHy and Stx7-SEP by replotting as a function of stimulus numbers clearly showed that the rise kinetics of Stx7-SEP were identical to those of SypHy during 20 and 40 Hz stimulation, whereas those of Stx7-SEP were much slower than SypHy at 5 and 10 Hz stimulation (Fig. 2d).

The reluctant, and incomplete recruitment of Stx7-SEP vesicles during prolonged stimulation at low frequencies (5 or 10 Hz) allows us to examine whether Stx7-SEP vesicles represent a subgroup of the total recycling pool labeled with SypHy, or whether they are separate groups, although TeNT treatment clearly abolished Stx7-SEP responses within a restricted time frame (Fig. 1g). To this end, neurons expressing either SypHy or Stx7-SEP were subjected to low-frequency stimulation at 5 Hz, for 500 APs, in order to deplete the total recycling pool of SypHy, and then subsequently subjected to HFS at 40 Hz, 600 APs (Supplementary Fig. 7). The second stimulation at 40 Hz produced a scant increase in SypHy fluorescence, whereas the same stimulation produced a drastic increase in Stx7-SEP, indicating that Stx7-SEP vesicles represent a negligible portion of the total recycling pool. However, pretreatment of neurons with TeNT did not completely abolish the Stx7-SEP response to the second stimulation at 40 Hz, incompatible with results in the absence of prolonged pre-stimulation at 5 Hz (Fig. 1g and Supplementary Fig. 2), strongly indicating that Stx7-SEP vesicles recruited for the release under this condition were not typical SVs. These unexpected observations can be explained if a substantial shift of Stx7-SEP from the SV pool to the non-SV pool occurs, perhaps mediated by fusion events via Stx7 and other endosomal SNAREs in non-SV compartments during the prolonged stimulation at 5 Hz. The unsolved question of whether Stx7-SEP vesicles comprise part of the total recycling pool will be addressed by other approaches, as described below (see Figs. 6 and 7).

**Stx7 localizes to a subset of SVs with low abundance at the presynaptic terminals.** In order to gain further evidence for the localization of Stx7 in a subpopulation of SVs, we adopted morphological and biochemical approaches. First, we asked

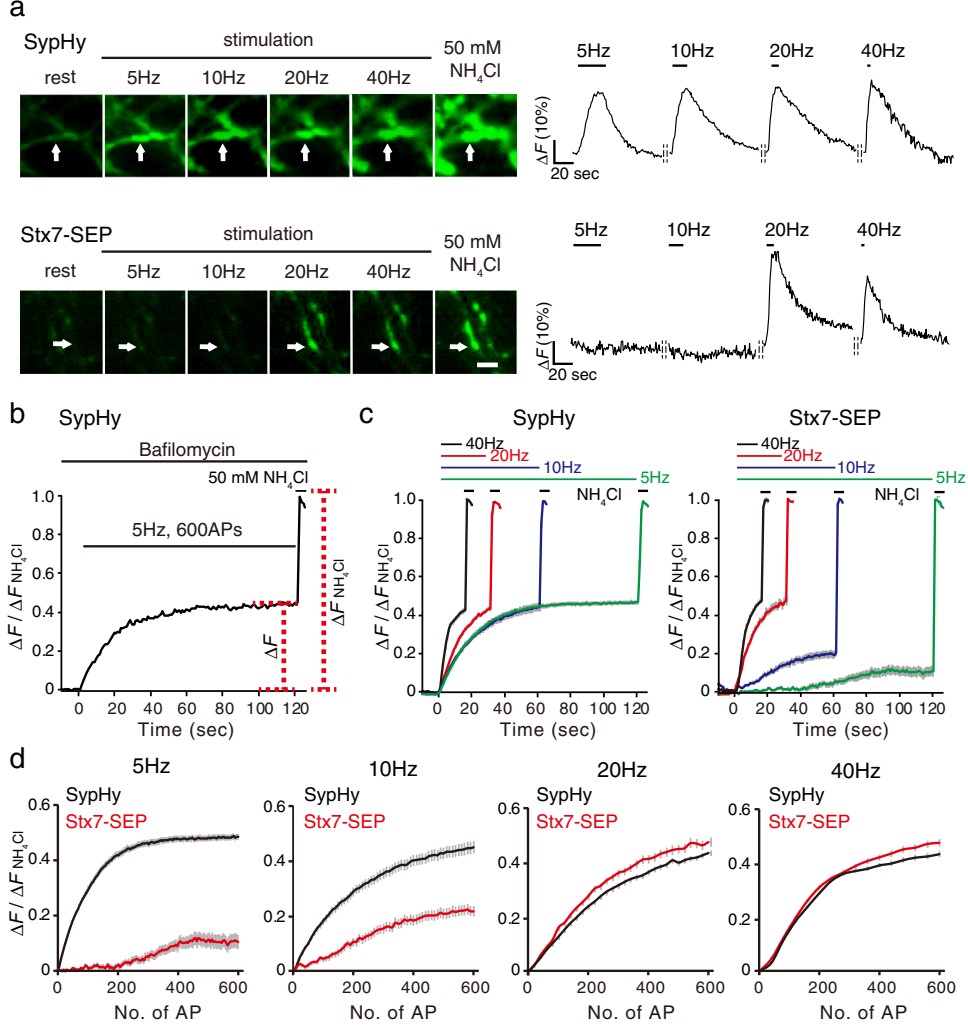

**Fig. 2 Stx7-SEP-vesicles recycle preferentially upon HFS. a** Stimulus-frequency-dependent responses of Stx7-SEP in comparison to SypHy. Neurons expressing respective SEP fusion constructs were subjected to sequential stimulation ranging from 5 to 40 Hz (200 APs) at 5 min intervals. Left images are representative images of SypHy (top), Stx7-SEP (bottom) at rest, at the end of stimulation at 5, 10, 20, and 40 Hz, and upon application of $NH_4Cl$ at the end of recordings. Right traces show representative traces of each SEP-fluorescence change in boutons, indicated by arrows. Data were normalized to fluorescence signals upon application of $NH_4Cl$ at the end of recordings. **b** Experimental scheme to estimate the kinetics of exocytosis, as well as sizes of total recycling SV pools. Neurons were pretreated with 2 μM bafilomycin A1 for 60 s and then stimulated with 600 APs at different stimulation frequencies. After cessation of stimulation, 50 mM $NH_4Cl$ was applied and fluorescence during $NH_4Cl$ application was used to normalized fluorescence signals at individual boutons. **c** Recycling pool of SypHy and Stx7-SEP. Cells expressing SypHy (left) and Stx7-SEP (right) were stimulated with 600 APs at different frequencies (5, 10, 20, and 40 Hz). **d** Replots of SypHy (black) and Stx7-SEP (red) responses of the results in (**c**) as a function of stimulus numbers (No. of AP).

whether endogenous Stx7 localizes to a specific part of the SV cluster. To this end, we co-stained cultured hippocampal neurons with antibodies against Stx7 and Syp, the latter of which should illuminate entire SV clusters. Although Stx7 immunoreactivity was apparent in cell somas and dendrites, it was also observed along axons (Supplementary Fig. 8a). Images at higher magnification revealed that Stx7 fluorescence partially overlapped with Syp signals (Fig. 3a). We then performed triple staining, including an active zone (AZ) marker bassoon (BSN). Whereas BSN signals often appeared as small, compact structures located at the center of Syp-positive puncta, Stx7 immunoreactivity only partially overlapped with BSN signals and located surrounding BSN signals (Fig. 3b and Supplementary Fig. 8b), indicating that intrinsic Stx7 localizes at the distal side of the SV cluster from the AZs. To substantiate these observations, we next performed immunoelectron microscopy. Since our initial attempts to detect endogenous Stx7 with the same antibody did not produce reliable

signals, we expressed Stx7-SEP as performed for SEP imaging, and proceeded to immunostaining using anti-GFP antibody. In comparison, we adopted cultured cells transfected with SypHy, which could be detected with the same antibody. Consistent with immunofluorescence data, immunoparticles for SypHy were widely spread all over SV clusters in presynaptic structures, whereas Stx7-SEP immunoparticles were sparsely distributed within SV clusters (Fig. 3c and Supplementary Fig. 9). Quantification of densities of immunoparticles revealed that SypHy was expressed at a significantly higher level than Stx7-SEP (approximately 6-fold; mean ± s.e.m., 294 ± 33 particles/μm² for SypHy vs. 47 ± 5 particles/μm² for Stx7-SEP), while the numbers of both immunoparticles showed a positive correlation with areas of synaptic varicosity (Fig. 3d, e and Supplementary Fig. 9). Notably, distances of Stx7-SEP immunoparticles to the nearest AZ membranes (defined by electron-dense postsynaptic density (PSD) structures) were significantly longer than those of SypHy, which

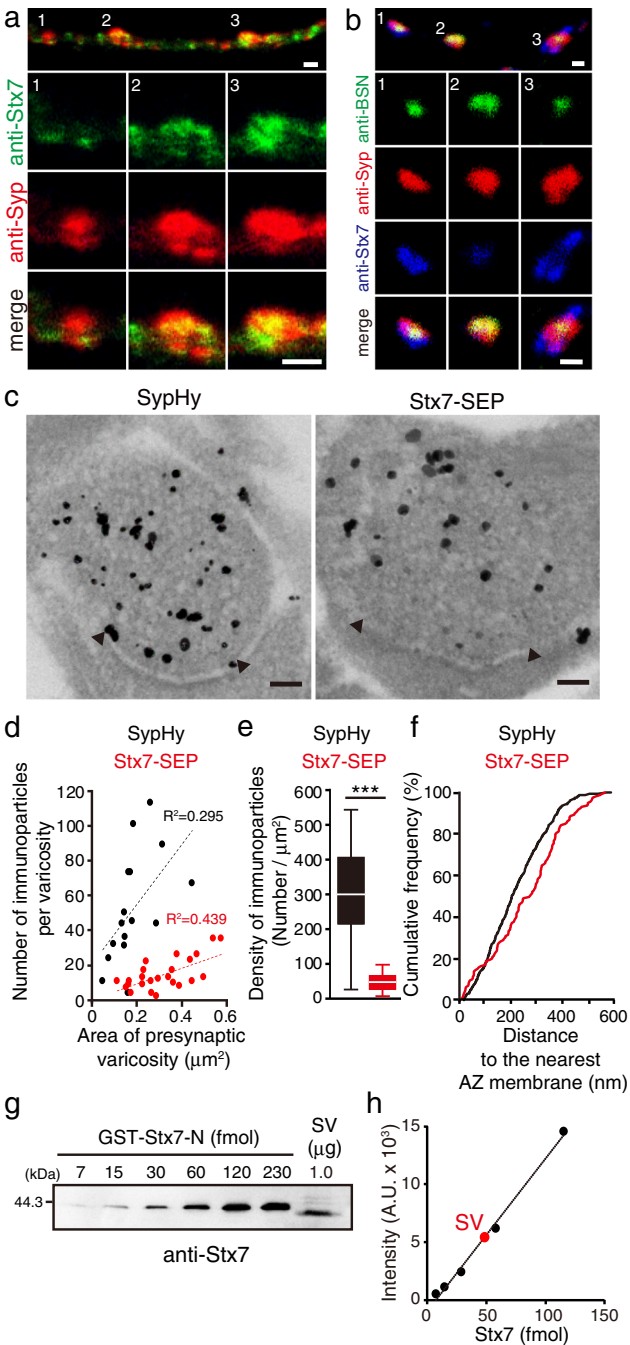

**Fig. 3 Stx7 localizes to a subpopulation of SVs at presynaptic terminals. a** Double immunostaining of Stx7 and Syp in cultured hippocampal neurons at 14 DIV. An upper panel shows representative axonal localization of Stx7 (green) and Syp (red). Magnified images of numbered areas (1–3) are shown individually below. The specificity of the Stx7 antibody was confirmed in independent experiments where Stx7 expression level was reduced by specific shRNA (Supplementary Fig. 14b). Scale bars indicate 2 µm. **b** Triple-immunofluorescence for Stx7 (blue), Syp (red), and an active zone maker Bassoon (BSN, green). An upper panel shows representative images. Magnified images of the numbered areas (1–3) are shown below. Scale bars indicate 1 µm. **c** Immunoelectron micrographs of SypHy (left) and Stx7-SEP (right) at presynaptic terminals. Immunogold labeling was intensified by silver enhancement. Arrowheads indicate both edges of the active zone deduced from postsynaptic density structures. Scale bars indicate 100 nm. **d** Number of immunoparticles as a function of the area of presynaptic varicosity. Sixteen varicosities for SypHy (black) and 24 varicosities for Stx7-SEP (red) were analyzed. **e** A box-whisker plot showing densities of SypHy immunoparticles (black) and Stx7-SEP immunoparticles (red) calculated from (**d**). $p < 0.0001$ with unpaired $t$-test with Welch's correction. **f** A cumulative plot of distances of immunoparticles to the nearest AZ membrane. Note that vertical lines from the edges of AZs were drawn manually, and only immunoparticles inside the enclosed areas were measured for the analysis (Supplementary Fig. 9). Statistical significance was evaluated with Kolmogorov–Smirnov test ($p = 0.0003$). **g** A representative quantification of Stx7 in native SVs purified from rat brains. Various amounts of recombinant GST-Stx7-N-terminal domain (GST-Stx7-N) and a fixed amount of purified SV fraction (vesicle concentration, 26.7 nM; protein concentration, 99.7 ng/µL) were subjected to quantitative western blot analysis (see also Supplementary Fig. 10 for complete datasets and control experiments for Syb2). **h** Signal intensities of bands were quantified and plotted as a function of moles of GST-Stx7-N. A red circle indicates the signal intensity measured for 1.0 µg SV shown in (**g**).

Stx7 molecules located on SVs is calculated to be ~53, which roughly coincides with the number of Stx7 molecules in the synaptosomal fraction[7], suggesting that more Stx7 molecules are present in presynaptic terminals than in postsynaptic compartments.

**Recruitment of Stx7-vesicles for exocytosis is triggered by the Ca²⁺/calmodulin pathway and requires actin polymerization.** The fact that Stx7-SEP responded preferentially to HFS (>20 Hz) suggested that exocytosis of Stx7-SEP-bearing SVs would require a high concentration of $Ca^{2+}$. To test this hypothesis, we raised external $Ca^{2+}$ from 2 to 8 mM, and examined whether even 10 Hz stimulation would cause robust responses. The response of SypHy was facilitated in the presence of 8 mM $Ca^{2+}$ (Fig. 4a, left), as observed previously[30]. Notably, Stx7-SEP also exhibited a robust exocytic response to 10 Hz stimulation when external $Ca^{2+}$ was raised to 8 mM (Fig. 4a, right). To further elucidate signaling pathways downstream of $Ca^{2+}$, we focused on calmodulin, since it is a $Ca^{2+}$-sensor protein that mediates fast SV replenishment after RRP depletion at the calyx of Held synapse and at hippocampal synapses[31,32]. To this end, we co-expressed the SEP reporters and a calmodulin inhibitory peptide (CIP), and monitored fluorescence changes in the presence of Baf for prolonged periods (either at 10 Hz for 60 s or at 20 Hz for 30 s). Consistent with previous reports, CIP expression slowed SV recruitment, measured with SypHy, during 20 Hz stimulation (Fig. 4b), whereas no changes were observed during 10 Hz stimulation (Fig. 4b), indicating that the activation of the $Ca^{2+}$/calmodulin pathway promotes rapid SV replenishment, preferentially during intensive stimulation. No changes in total pool sizes were observed at either stimulation frequency (Fig. 4b).

was evident from a right shift of a cumulative plot for Stx7-SEP compared to that for SypHy (Fig. 3f). These observations were fully compatible with our immunofluorescence data, as well as with a previous observation under STED microscopy[7].

We next attempted to determine the copy number of Stx7 in an SV. Previously, Stx7 content in the synaptosomal fraction isolated from rat brains was reported to be 78.6 copies per synapse[7]. However, the synaptosomal fraction also contains postsynaptic compartments in which Stx7 might be present (Supplementary Fig. 8a). To estimate a fractional contribution of presynaptic Stx7 to the synaptosomal fraction, we quantified the Stx7 content in purified SVs in which Stx7 was shown by western blot analysis to be highly enriched[6]. We estimated the copy number of Stx7 molecules per SV to be $0.14 \pm 0.03$ (mean ± standard deviation) (Fig. 3g, h and Supplementary Figs. 10, 11). Assuming that an average synaptosome contains ~380 SVs[7], the total number of

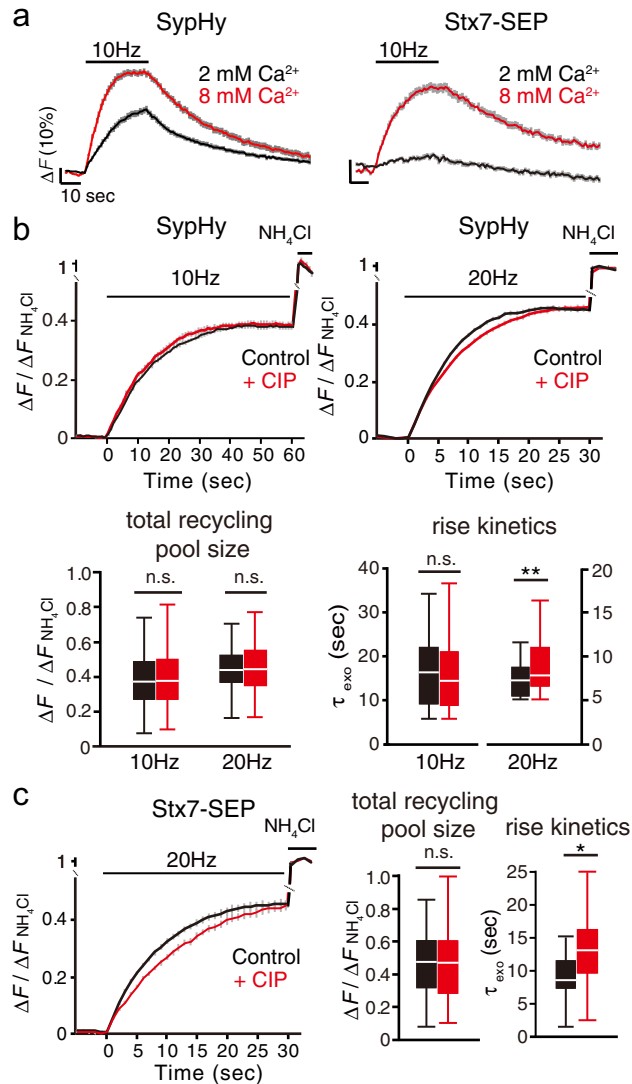

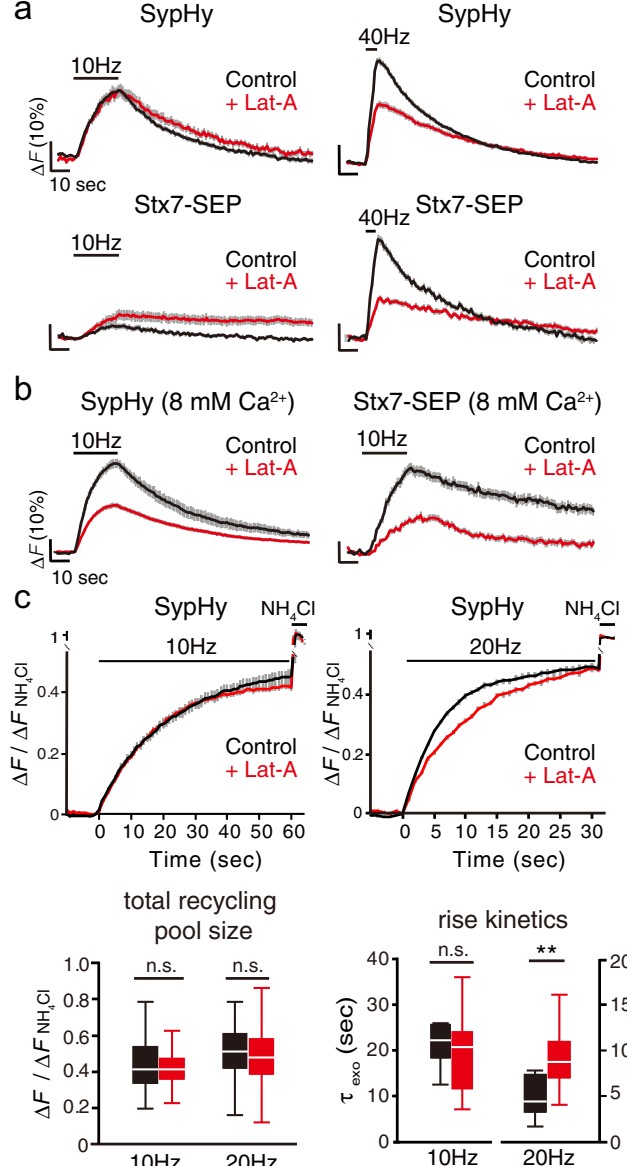

**Fig. 4 Fast recruitment of Stx7-vesicles is mediated by activation of the Ca²⁺/calmodulin pathway. a** Responses of Stx7-SEP (right) in the presence of normal (2 mM, black) and high (8 mM, red) external Ca²⁺. For comparison, responses of SypHy under identical conditions are shown in the left panel. **b** Effects of CIP (red) on exocytosis of total recycling pool monitored by SypHy. Responses at 10 Hz, 600APs (left) and 20 Hz, 600APs (right) with bafilomycin treatment are shown. Control experiments without CIP in respective conditions are shown in black. Bottom box-whisker plots show quantitative comparisons of total recycling pool sizes and rise kinetics. Normalized fluorescence peaks (left) or the time constants of rise time ($\tau_{exo}$) (right) during 10 Hz or 20 Hz stimulation are compared. **c** Effects of CIP (red) on exocytosis of the total recycling pool monitored by Stx7-SEP. Responses at 20 Hz, 600APs (left) with bafilomycin treatment are shown. Control experiments without CIP in the respective conditions are shown in black. Right box-whisker plots show quantitative comparisons of total recycling pool sizes and rise kinetics. Normalized fluorescence peaks (left) or the time constants of rise time ($\tau_{exo}$) (right) during 20 Hz stimulation are compared. All traces are average traces from >150 boutons.

**Fig. 5 Recruitment of Stx7-vesicles requires actin polymerization. a** Effects of latrunculin A (Lat-A, 5 μM; red) on responses of SypHy (upper traces) and of Stx7-SEP (lower traces) elicited by 10 Hz (left) and 40 Hz (right) stimulation. Control experiments without Lat-A in the respective conditions are shown in black. **b** Effects of Lat-A (red traces) on recycling of SypHy and Stx7-SEP in the presence of 8 mM external Ca²⁺. Control experiments without Lat-A in the respective conditions are shown in black. **c** Effects of Lat-A (red) on exocytosis of total recycling pool monitored by SypHy. Responses at 10 Hz, 600APs (left) and 20 Hz, 600APs (right) with bafilomycin treatment are shown. Control experiments without Lat-A in the respective conditions are shown in black. Bottom box-whisker plots show quantitative comparisons of total recycling pool sizes and rise kinetics. Normalized fluorescence peaks (left) or the time constants of rise time ($\tau_{exo}$) (right) during the 10 Hz or 20 Hz stimulation are compared. All traces are average traces from 50 to 150 boutons.

Notably, recruitment of Stx7-SEP vesicles for exocytosis was also significantly slowed by CIP at 20 Hz stimulation (Fig. 4c). These results indicate that a calmodulin-dependent SV replenishment process preferentially operates at HSF (>20 Hz) in cultured hippocampal neurons, and that Stx7-SEP-laden vesicles may well constitute (at least a part of) such an SV pool.

We next focused on the actin cytoskeleton, because Ca²⁺/calmodulin-dependent fast SV replenishment after RRP depletion reportedly requires actin polymerization in calyx of Held synapses[33,34]. In accordance with a previous report[9], actin depolymerization with latrunculin A (Lat-A: 5 μM) did not show remarkable effects on SypHy recycling during 10 Hz stimulation

(Fig. 5a). However, Lat-A significantly retarded the SypHy response upon 40 Hz stimulation (Fig. 5a, upper right), indicating that there exists a subset of SVs for which mobilization for release preferentially during the intense stimulation is actin-dependent. This result is consistent with previous observations obtained using neurons in which actin isoforms were conditionally delepted[33]. Intriguingly, more pronounced reduction by Lat-A was observed for Stx7-SEP responses upon 40 Hz stimulation (Fig. 5a), indicating that Stx7-SEP localizes to a subpopulation of recycling SVs for which recruitment for exocytosis is actin-dependent. Lat-B, another actin polymerization inhibitor, also decreased the SypHy and Stx7-SEP responses to a similar extent at 40 Hz, but not at 10 Hz (Supplementary Fig. 12). Finally, when Stx7-SEP and SypHy were monitored during 10 Hz stimulation in the presence of 8 mM $Ca^{2+}$, Lat-A significantly reduced the responses of both (Fig. 5b).

To further determine whether Lat-A treatment reduced the size of the total recycling pool or whether it simply slowed SV recruitment for release, we monitored SypHy fluorescence in the presence of Baf (Fig. 5c). Whereas release kinetics ($\tau_{exo}$) and the total recycling pool size of SypHy during 10-Hz stimulation were not affected by Lat-A, rise kinetics of SypHy during 20 Hz stimulation were significantly slowed by Lat-A with no changes in the total recycling pool size of SypHy (Fig. 5c).

Taken together, these results indicate that a portion of the SV recycling pool is recruited through a mechanism requiring activation of the $Ca^{2+}$/calmodulin pathway and actin polymerization, and that Stx7-SEP preferentially localizes to that pool, resembling a rapidly replenishing pool observed in calyx of Held synapses[31,33,35–37].

**The N-terminal domain of Stx7 is responsible for its characteristic behavior**. Like other Syntaxin family members, Stx7 comprises an N-terminal domain (NTD), the SNARE domain, and a transmembrane domain (TMD)[5] (Fig. 6a). To understand the molecular mechanism by which Stx7 is selectively sorted to a subset of the SV recycling pool, we constructed two deletion mutants either lacking the NTD (Stx7-ΔNTD-SEP) or the SNARE motif (Stx7-ΔSNARE-SEP) (Fig. 6a). When these mutants were expressed in cultured neurons, Stx7-ΔNTD-SEP was properly sorted to Syb2-positive bouton-like puncta (Fig. 6b) in which it is preferentially localized to the intracellular acidic compartments, with albeit higher luminal pH compared to genuine SVs (Fig. 6c, d and Supplementary Fig. 12). However, unlike Stx7-SEP, Stx7-ΔNTD-SEP responded to 10 Hz stimulation as it did to 40 Hz stimulation (Fig. 6e) and TeNT treatment did not completely abolish the responses at 10 Hz (Fig. 6f). Furthermore, responses of Stx7-ΔNTD-SEP at 40 Hz were no longer sensitive to Lat-A treatment (Fig. 6g). Thus, the NTD is essential for proper sorting of Stx7 to the actin-dependent SV subpool that is preferentially recruited during HFS. By contrast, Stx7-ΔSNARE-SEP did not accumulate at Syb2-positive bouton-like puncta, and was distributed evenly on cell surfaces (Fig. 6b–d and Supplementary Fig. 13), suggesting that the SNARE domain is indispensable for its presynaptic, as well as vesicular, localization.

**Overexpression of Stx7-NTD selectively reduces SV recruitment during HFS**. Although the results described above demonstrate that Stx7-SEP is directed to a subset of the SV recycling pool, whether Stx7 is necessary for this subpopulation of SVs remains unclear, especially given that only a small population of SVs carries Stx7 (Fig. 3). To address this question, we first examined whether silencing of Stx7 with specific shRNAs affects the SV recycling subpool. However, chronic knockdown of Stx7 expression severely reduced SypHy responses at 10 and 40 Hz,

indicating that Stx7 is necessary to establish this SV pool per se during synapse development and maturation processes (Supplementary Fig. 14). As an alternative approach to inactivate Stx7, we explored specific dominant-negative effects by overexpressing the N-terminal domain of Stx7. The rationale for this is that the NTD of Stx7 interacts with its own SNARE motif, thereby inhibiting SNARE complex formation with cognate SNAREs[37]. In addition, the results above (Fig. 6) clearly demonstrate that the NTD of Stx7 is essential for its proper sorting to the actin-dependent SV recycling pool. To this end, we placed a P2A self-cleaving peptide between the SypHy and Stx7-NTD sequences (a.a. 13-149) to ensure co-expression of both proteins[20], and monitored SV recycling reported by SypHy (Fig. 7a). Expression of Stx7-NTD did not alter the total recycling pool size or the kinetics of vesicle release elicited at 10 Hz (Fig. 7b). In stark contrast, expression of the Stx7-NTD significantly slowed the increase of SypHy fluorescence during 20 Hz stimulation, while total recycling pool size was affected only slightly (Fig. 7b). Treatment with Lat-A in addition to Stx7-NTD overexpression did not further slow release kinetics (Fig. 7c), indicating direct involvement of Stx7 in the actin-dependent SV recycling pool that rapidly and preferentially replenishes RRP upon HFS.

## Discussion

Departing from comprehensive analyses of optical reporters for multiple presynaptic endosomal SNAREs, we report that an endosomal Q-SNARE, called Stx7, specifically targets a unique SV pool that replenishes RRP only during intense activity. Recruitment of this SV pool, enabled by Stx7, is activated by the $Ca^{2+}$/calmodulin pathway and requires actin polymerization, characteristics resembling a population of the recycling SV pool that rapidly replenishes RRP in calyx of Held synapses[33,36–38]. Our results collectively indicate that endosomal fusion supported by Stx7, presumably together with its cognate SNAREs, is responsible for generating a functionally distinct recycling pool of SVs.

**Recycling vesicle pool heterogeneity conferred by endosomal SNAREs**. Previous studies using SEP constructs have suggested that release properties of a subpopulation of SVs are dependent upon endosomal SNAREs[9–12]. For instance, the Qc-SNARE, vti1a, and the R-SNARE, VAMP7, reside mainly on the reserve pool that is reluctantly mobilized during activity. Instead, SVs bearing these SNAREs are prone to fuse spontaneously in the absence of activity[9,10]. Likewise, other endosomal SNAREs, Stx6, Stx12/13, and VAMP4, are capable of recycling, albeit to a lesser extent than authentic SV proteins, such as Syb2[11,12]. However, our comprehensive SEP imaging analyses to examine effects of TeNT on stimulus-dependent fluorescence changes, as well as vesicular pHs of vesicle compartments on which they reside, reveal that presynaptic endosomal SNAREs can be categorized into two distinct classes; i.e., Stx7 and other endosomal SNAREs. First, Stx7-SEP resides on a fraction of the recycling vesicle pool that is preferentially mobilized during HFS (e.g., >20 Hz). Mobilization of this pool depends on an increase in cytosolic $[Ca^{2+}]$ exceeding a certain threshold to activate calmodulin, and also requires actin dynamics (Figs. 4 and 5). Since Stx7-laden vesicles share some characteristic features of genuine SVs, e.g., the same vesicular pH as SypHy- or VGLUT1-carrying vesicles, and complete silencing of activity-dependent exocytosis by TeNT treatment, we conclude that Stx7 resides on a subpopulation of the recycling SV pool.

In great contrast to Stx7, other presynaptic endosomal SNAREs exhibit different characteristics and behaviors compared to genuine SV residents. While they responded to a wide range of stimulation, their exocytotic responses were only partially

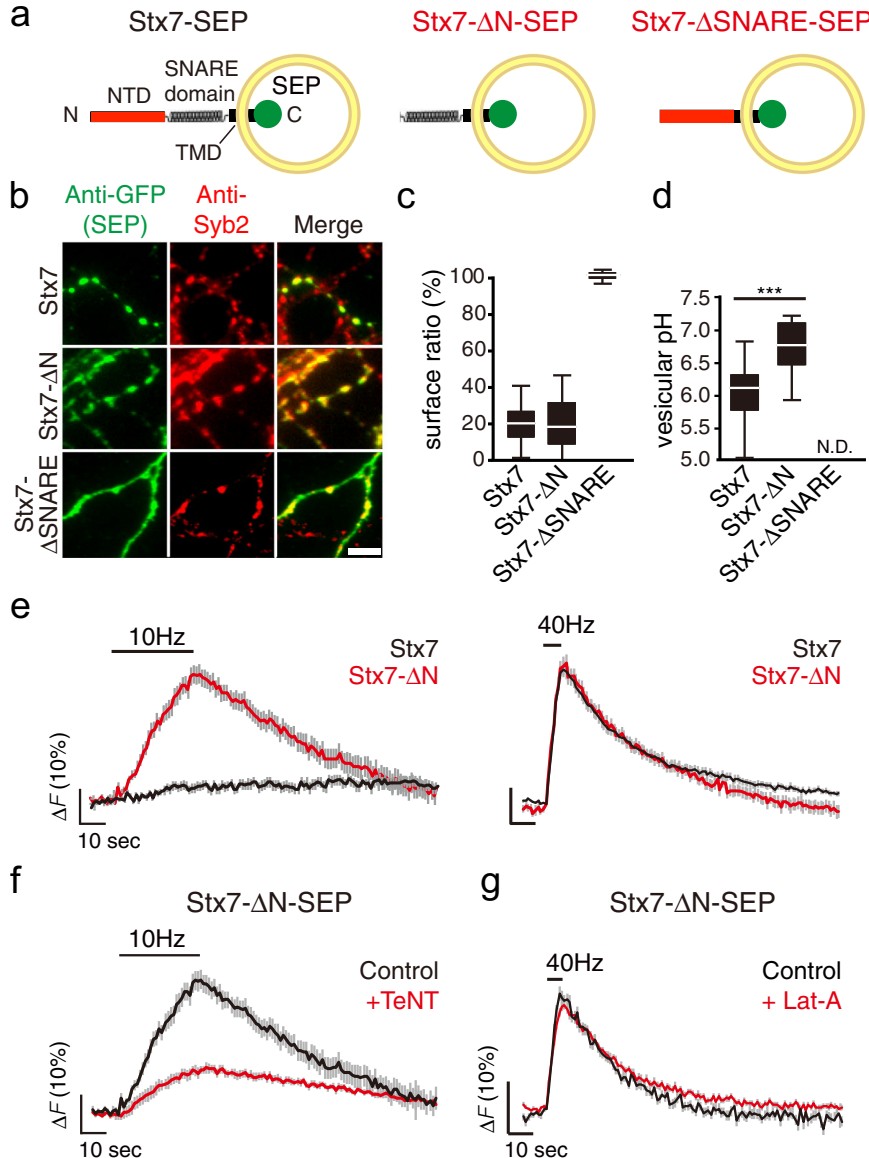

**Fig. 6 The N-terminal domain of Stx7 is responsible for proper sorting of Stx7 to a subset of recycling SVs. a** Schematic diagram of full length Stx7-SEP, Stx7-SEP lacking the N-terminal domain (Stx7-ΔN-SEP), and Stx7-SEP lacking SNARE motif (Stx7-ΔSNARE-SEP). **b** Distribution of Stx7-SEP, Stx7-ΔN-SEP, and Stx7-Δ-SNARE-SEP in transfected neurons. Box-whisker plots showing the surface fraction of Stx7-SEP and truncated mutants (**c**) and vesicular pHs of vesicles carrying respective SEPs (**d**). Note that vesicular pH of Stx7-ΔSNARE-SEP could not be calculated, since it was exclusively expressed at the cell surface (N.D. indicates 'not determined'). **e** Responses of Stx7-SEP (black traces) and Stx7-ΔN-SEP (red traces) upon 10 Hz (left panels) and 40 Hz stimulation (right panels). **f** Responses of Stx7-ΔN-SEP upon 10-Hz stimulation in control (black) and after TeNT treatment (red). **g** Responses of Stx7-ΔN-SEP upon 40 Hz stimulation in control (black) and after Lat-A treatment (red).

attenuated by TeNT treatment, indicating that these endosomal SNAREs are present, albeit to various extents, on Syb2/VGLUT1-negative vesicle compartments. This is further supported by the fact that the average vesicular pH of vesicles containing these endosomal SNAREs appeared to be significantly higher (~6.4) than the pH of authentic SV residents. Thus, although it was proposed that Stx6, Stx12/13, and vti1a are involved in the replenishment of RRPs[11], they are only minimally present in the RRP and recycling SV pool. Specifically, they are present in non-SV compartments, which nevertheless recycle in an activity-dependent manner. Additionally, although a previous study proposed that vti1a selectively resides on Syb2-free vesicles, which are preferentially utilized for spontaneous release rather than stimulus-dependent evoked release[10], our results reveal that the vast majority of presynaptic endosomal SNAREs are present on Syb2-free vesicles. The identity and function of such non-SV recycling vesicles at presynaptic terminals conferred by these endosomal SNAREs are enigmatic, but they may utilize non-canonical neuronal SNARE proteins and a $Ca^{2+}$ sensor for exocytotic fusion with the plasma membrane, such as SNAP-29 or -47, Stx3 or 4, and Synaptotagmin-7[32,38,39]. Notably, recent studies have proposed the existence of non-SV-type secretory organelles at synaptic sites or in close proximity to presynaptic active zones, which secrete neuropeptides from lysosomes or dense-core vesicles[38,40,41]. Unraveling SNAREs or $Ca^{2+}$ sensors involved in these non-SV vesicles will enable us to understand the physiological significance of these non-SV secretory organelles for synaptic performance and signaling.

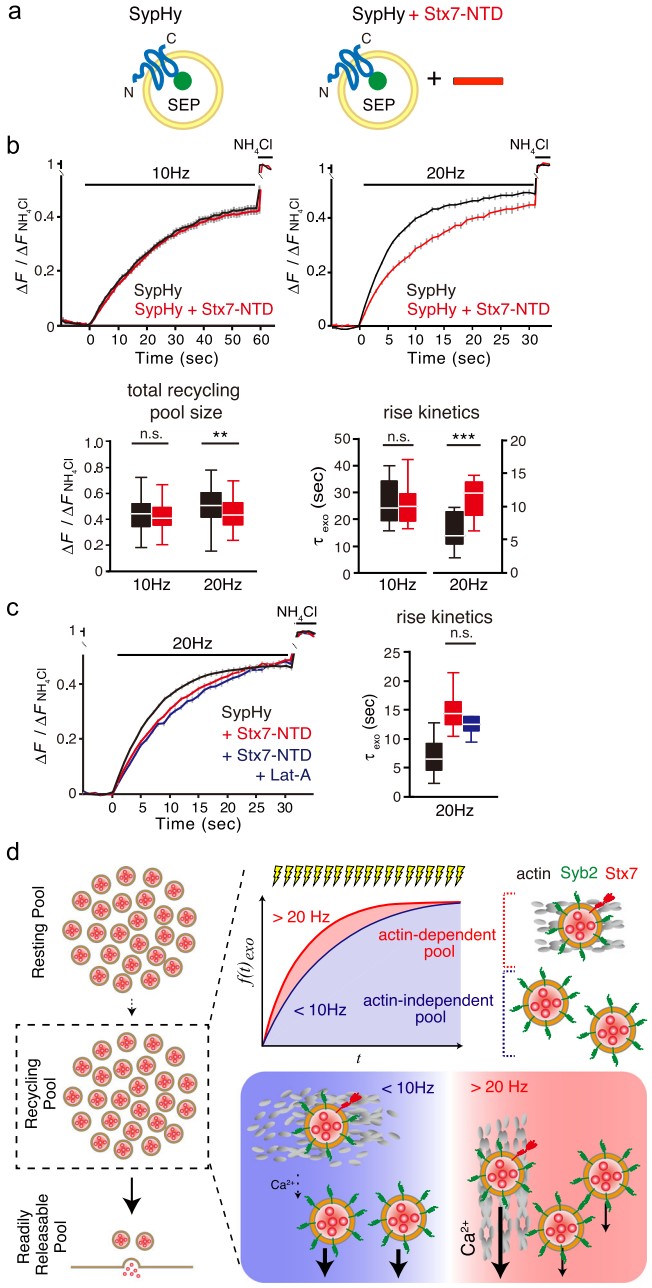

**Fig. 7 Overexpression of Stx7-NTD slowed SypHy responses only during HFS. a** Schematic diagram of SypHy and SypHy co-expressed with Stx7-NTD. Stx7-NTD was placed after a P2A sequence so that all SypHy-positive cells co-expressed Stx7-NTD. **b** The total recycling pool monitored by SypHy responses in the absence (black) and presence (red) of Stx7-NTD at 10 Hz, 600APs (left) and 20 Hz, 600APs (right) after bafilomycin treatment. Bottom box-whisker plots show quantification of total recycling pool sizes (left) and time constants of rise time (right, $\tau_{exo}$). **c** Pretreatment with Lat-A did not further reduce release kinetics upon Stx7-NTD overexpression. Responses of SypHy with Stx7-NTD in the presence of Lat-A (blue) were compared with control (SypHy only, black) and SypHy with Stx7-NTD (red) in response to at 20 Hz, 600APs. A box-whisker plot shows time constants of the rise time ($\tau_{exo}$). SypHy responses with Stx7-NTD (red) and Stx7-NTD + Lat-A (blue) did not differ significantly ($p > 0.05$). **d** Schematic summary of this study, depicting that Stx7 is preferentially present in the rapidly replenishing SV pool during intense, repetitive stimulation.

**Actin and SV mobilization.** The role of the actin cytoskeleton in regulating synaptic vesicle mobilization has long been controversial. While Lat-A, an actin polymerization inhibitor, had little impact on SV mobility or recycling in hippocampal neurons[9,42], similar treatment resulted in a deceleration of SV replenishment of the RRP in calyx of Held synapses[35–37], indicating the presence of actin-dependent, rapidly replenishing SVs. Here, using SypHy imaging, we provide evidence compatible for both views regarding the role of actin dynamics in SV exocytosis and recruitment. Specifically, SVs are recruited for release in an actin-independent manner during mild stimulation, while the exocytotic release of a significant fraction of SVs is accelerated by an actin-dependent process during high-frequency stimulation (Fig. 5). The former is consistent with the previous observations[9,42] and the latter is compatible with reduced SV replenishment into the RRP proposed in calyx of Held synapses, reportedly mediated by $Ca^{2+}$/Calmodulin-dependent processes[31]. Interestingly, SVs carrying Stx7-SEP are preferentially utilized for release during intense activity and are more dependent on actin dynamics than the remainder of the recycling pool revealed by SypHy. Thus, our results collectively indicate that SVs characterized by Stx7 may represent a fraction of the recycling SV pool rapid mobilization of which for release requires actin polymerization, presumably mediated by the $Ca^{2+}$/calmodulin pathway, to replenish the RRP during high activity.

**How does Stx7 define rapidly replenishing SVs?** How does Stx7 contribute to rapidly replenishing SVs? In non-neuronal cells, Stx7 is required for homotypic/heteromeric fusion of late-endosomes and lysosomes[26,27,43] or lysosomes and autophagosomes[44], and these processes may contribute to protein sorting and organelle maturation[45]. In accordance with the role of Stx7 in non-neuronal cells, we propose that Stx7 may contribute to the formation of the rapidly replenishing SV subpool by mediating homotypic, as well as heterotypic fusion with existing endosomal structures. Recent evidence suggests that the generation of clathrin-independent, large, endosome-like vacuoles or bulk endocytosis is the predominant form of SV endocytosis, especially after repetitive stimulation[46–48]. Although current models predict clathrin-mediated SV reformation directly from these endosomal membranes, it is also evident that homotypic or heterotypic fusion of endosomes mediated by endosomal SNAREs can occur before SV reformation, even with multiple rounds during high activity[49,50]. Although a molecular link between Stx7 and actin is enigmatic, Stx7 may help generate the rapidly replenishing SV pool, as a result of endosomal fusion and molecular sorting. This model seems to work if Stx7, but not other cognate SNAREs, are destined to SVs, and if exogenously expressed Stx7-SEP follows the trafficking route of endogenous Stx7.

Some other scenarios may explain the contribution of Stx7 to the observed results. First, it is possible that Stx7 directly or indirectly associates with actin filaments and that rapid recruitment of Stx7-vesicles is only possible when the $Ca^{2+}$/calmodulin cascade is activated. Alternatively, Stx7 may contribute to compound membrane fusion with its cognate SNAREs, involving the fusion of multiple vesicles prior to a final fusion[51]. However, considering the limited number of copies of Stx7 in a subpopulation of SVs (only 14% of SVs can harbor a single copy of Stx7) and the relatively high impact of Stx7-NTD overexpression on the SypHy response, as well as the highly specific behavior of Stx7 among presynaptic endosomal SNAREs, these are not likely possibilities. Further studies will be necessary to determine how Stx7, presumably with its cognate SNAREs, functionally differentiates this SV pool, thereby affecting activity-dependent synaptic performance.

## Methods

**Molecular biology**. SypHy, Syp-mOr, VGLUT1-SEP, and endosomal SNARE-SEPs were expressed in a neuron-specific manner using lentivirus-based vectors, in combination with the Tet-Off system[19,20]. Two vectors were used, a 'regulator' vector expressing an advanced tetracycline transactivator (tTAad) under the control of the human synapsin 1 promoter (pLenti6PW-STB), and a 'response' vector that expressed SypHy, Syp-mOr, VGLUT1-SEP or endosomal SNARE-SEPs under the control of a modified tetracycline-response element (TRE) composite promoter (pLenti6PW-TRE). To construct endosomal SNARE-SEPs and deletion mutants, mouse coding sequences for Stx6 (accession no. NM_021433.3), Stx7 (accession no. NM_016797.4), Stx8 (accession no. NM_018768.2), Stx12/13 (accession no. NM_018768.2), Stx16C (accession no. NM_001102424.1), vti1a (accession no. NM_001293685.1), VAMP4 (accession no. NM_016796.3), VAMP7 (accession no. NM_001302138.1), vti1b (accession no. NM_006370.3), VAMP8 (accession no. NM_016794.3), Stx7-ΔN (a.a.168-261) and Stx7-ΔSNARE (a.a.1-167, 228-261) without stop codons were amplified by PCR using mouse adult brain complementary DNA as a template, and fused with SEP. Thereafter, they were cloned into a SmaI site located downstream of the TRE sequence in pLenti6PW-TRE using an In-Fusion Cloning Kit (Clontech)[19,20].

VGLUT1-SEP was constructed as described previously[18] and cloned into the SmaI site of pLenti6PW-TRE. SypHy and Syp-mOr plasmids were generated as described previously[19,20].

To co-express SypHy and the N-terminal domain of Stx7 (Stx7-NTD; a.a.13-137), a sequence encoding a self-cleaving P2A peptide was placed between SypHy and Stx7-NTD[20]. To express calmodulin inhibitory peptide (CIP; a.a. 2162-2178 of Xenopus laevis myosin light chain kinase, accession no. XP_018091115), tagRFP or tagRFP-P2A-CIP was amplified by PCR from ptagRFP-C (evrogen), and cloned into AgeI and EcoRI site of FUGW, a gift from David Baltimore (Addgene plasmid # 14883; http://n2t.net/addgene:14883; RRID:Addgene_14883)[52].

To generate recombinant proteins as standards for quantification analysis, Stx7-NTD (a.a. 1-149) and Syb2-ΔTMD (a.a. 1−94) (accession no. NM_009497.3) were amplified by PCR, and cloned into the BamHI site and SalI site of pGEX6P-1. Lentiviral vectors carrying mCherry and either scrambled shRNA (CCTAAGGTTAAGTCGCCCTCG), mouse Stx7 shRNA-#1 (CGATATGATTGACAGCATAGA), or mouse Stx7 shRNA-#2; (GAAGCTGATATTATGGACATT) were obtained from VectorBuilder Biotechnology Co. Ltd.

**Lentiviral vector production**. Lentiviral vectors to express SEP or tagRFP constructs, as well as shRNAs for Stx7 in cultured neurons, were produced in the human embryonic kidney (HEK) 293T cells, as described previously[19,20]. Briefly, HEK293T cells cultured in a 75 cm² flask (Falcon) at ~50% confluency were transfected with 17 μg lentiviral backbone plasmids (either pLenti6PW-STB, pLenti6PW-SEP reporters, FUGW-tagRFPs or shRNA vectors) and helper plasmids (10 μg pGAG-kGP1, 5 μg pCAG-RTR2, and 5 μg pCAG-VSVG) using a calcium phosphate transfection method[53]. 16−24 h after transfection, the culture medium was replaced with 7 mL of fresh Neurobasal medium supplemented with 2% B27 and 0.5 mM glutamine. Supernatants were collected 48 h later, and filtered through a 22 μm filter unit to remove cell debris. Aliquots were flash-frozen in liquid nitrogen and stored at −80 °C until use.

**Neuronal culture and viral expression**. Primary hippocampal cultures were prepared from embryonic day 16 ICR mice, as described previously[19,20], with slight modifications. Briefly, hippocampi were dissected and incubated with papain (90 units/mL, Worthington) for 20 min at 37˚C. After digestion, hippocampal cells were plated onto poly-D-lysine-coated coverslips framed either in 24 well plates, 12 well plates (Falcon), or a Nunc four-well dish (Thermo Fisher) at a cell density of 20,000−30,000 cells/cm², and kept in a 5% CO₂ humidified incubator. At 2−4 days in vitro (DIV), 40 μM FUDR (Sigma) and 100 μM uridine (Sigma) were added to inhibit the growth of glial cells. One-fifth of the culture medium was routinely replaced with fresh medium every 2−4 days until recordings. Cultures were transduced with two lentiviral vectors, the regulator vector and response vectors that encode SEP reporters or shRNAs for Stx7 at between 2 to 7 DIV, and were subjected to experiments at 12−14 DIV.

**Live imaging**. Fluorescence imaging was carried out at room temperature (~27 °C) on an inverted microscope (Olympus) equipped with a 60× (1.35 NA) oil immersion objective and a 75 W Xenon lamp. Cells cultured on glass coverslips were placed in a custom-made imaging chamber on a movable stage and continuously perfused with a standard extracellular solution containing (in mM): 140 NaCl, 2.4 KCl, 10 HEPES, 10 glucose, 2 CaCl₂, 1 MgCl₂, 0.02 CNQX, and 0.025 D-APV (pH 7.4). Field stimulation at various frequencies from 5 to 40 Hz (indicated in each figure) was delivered via bipolar platinum electrodes with 1 ms constant voltage pulses (50 V). At the end of recordings, an extracellular solution containing (in mM): 50 NH₄Cl, 90 NaCl, 2.4 KCl, 10 HEPES, 10 glucose, 2 CaCl₂, and 1 MgCl₂ (pH 7.4) was applied directly onto the area of interest with a combination of a fast flow exchange microperfusion device and a bulb controller, both of which were controlled by Clampex 10.2. To measure the kinetics of exocytosis and total recycling pool size, neurons were treated with a standard extracellular solution containing 2 μM bafilomycin A1 (Baf) (BioViotica) for 1 min, and placed

in the imaging chamber. Baf-containing solution was continuously applied to neurons throughout the recordings. For HFS, we adopted a stimulation frequency of 20 Hz instead of 40 Hz, since prolonged stimulation at 40 Hz often causes unavoidable air bubbles around stimulation electrodes, which hindered reliable imaging. For TeNT experiments, neurons were incubated with 10 nM recombinant TeNT (Sigma-Aldrich) for 16 h in a CO₂ incubator. For Lat-A and Lat-B experiments, neurons were treated with either 5 μM Lat-A (Wako) or 10 μM Lat-B (Cayman Chemical) for 1 min at RT, and subjected to imaging experiments. Lat-A- or Lat-B-containing buffer was continuously supplied to neurons throughout the recordings.

To estimate luminal pH and the surface fraction of SEP probes, a low pH solution containing (in mM): 140 NaCl, 2.4 KCl, 10 MES, 4 MgCl₂ (pH 5.5), and 50 mM NH₄Cl solution described above were successively applied to cultured neurons as described previously[19,20]. To restrict analyses at active synaptic boutons, electrical stimulation at 40 Hz was applied for 10 s at the end of recordings.

Fluorescence images (1024 × 1024 pixels) were acquired with an ORCA-Flash 4.0 sCMOS camera (Hamamatsu Photonics) in time-lapse mode either at 1 Hz (for imaging in response to stimulation) or 5.7 Hz (for estimating vesicular pH and surface fraction) under control of MetaMorph software (Molecular Devices). Fluorescence of SypHy, VGLUT1-SEP, or endosomal SNARE-SEPs was imaged with 470/22 nm excitation and 514/30 nm emission filters, and Syp-mOr was imaged with 546- to 556 nm excitation and 575- to 625 nm emission filters.

**Image data analysis**. Acquired fluorescence images were analyzed using Meta-Morph software. Active synapses were identified manually by changes in live measurements of fluorescence. Circular regions of interest (ROIs; 2.26 μm diameter and 4 μm² area) were positioned manually at the center of the highlighted fluorescence spots. Non-responsive puncta, especially for presynaptic endosomal SNARE-SEPs, which became apparent only upon NH₄Cl application, were excluded from analysis, since they may represent endosomal SNARE-SEPs expressed outside of presynaptic terminals. To extract fluorescence changes associated with the stimulation at each bouton, an average of five consecutive fluorescence values was taken as $F_0$, and this value was subtracted from subsequent fluorescence values (ΔF). To estimate relative fluorescence over total SEP molecules, the peak value during NH₄Cl application was taken as $\Delta F_{NH_4Cl}$ ($F_{NH_4Cl} − F_0$) and used to normalize the data ($\Delta F/\Delta F_{NH_4Cl}$). For each construct, fluorescence changes of at least 50 boutons were analyzed. To calculate rise kinetics of the fluorescence increase ($\tau_{exo}$), an average trace from one image (containing 5–10 active boutons) was collected, and taken as $n = 1$. For each averaged trace, $\tau_{exo}$ was calculated by fitting the trace with a single exponential function using a Solver function from Excel software. Time constants obtained from at least ten images were statistically evaluated.

Acquired images for vesicular pH and surface fraction measurements were also processed as described above. Vesicular pH and surface fraction were calculated as described previously[19,20], except that fluorescence during NH₄Cl application was approximated as fluorescence at pH 7.4. Values of pK$_a$ (7.1) and a Hill co-efficient (1.0) for SEP reported previously[16,25] were used for calculation.

**Immunocytochemistry**. Cultured neurons at DIV12–14 were fixed with 4% (wt/vol) paraformaldehyde in phosphate buffer (Wako) for 10 min at RT. After washing with PBS, cells were permeabilized with PBS containing 0.2% Triton X-100 for 20 min at RT, and incubated with PBS containing 10% (vol/vol) FBS for 30 min at RT. Cells were incubated with rabbit polyclonal anti-GFP antiserum and mouse monoclonal anti-Syb2 antibody (Cl69.1) (kind gifts from Reinhard Jahn) (Figs. 1b, 6b and Supplementary Fig. 4a), or with rabbit polyclonal anti-Stx7 antibody (Bethyl Laboratories; A304-512A), mouse monoclonal anti-Synaptophysin antibody (Cl7.2), guinea pig polyclonal anti-Synaptophysin antibody (Synaptic Systems; 101 004), and mouse monoclonal anti-Bassoon antibody (Novus Biologicals; SAP7F407)(Fig. 3a, b), or chick polyclonal anti-MAP2 antibody (1:1,000; Abcam; ab5392) and anti-Stx7 antibody (Supplementary Figs. 8a, 14b), for 1 h at RT. Cells were rinsed 3× with PBS, and further incubated with Alexa-488-conjugated anti-rabbit IgG (1:1,500; Invitrogen), Alexa-555-conjugated anti-mouse IgG (1:1,500; Invitrogen), Alexa-633-conjugated anti-guinea pig IgG (1:1,500; Invitrogen), or Alexa-633-conjugated anti-chick IgG (1:1,500; Invitrogen) for 30 min at RT. Fluorescence images were acquired with an inverted microscope (Olympus) with an ORCA-Flash 4.0 sCMOS camera (Hamamatsu Photonics) irradiated by a 75 W Xenon lamp (Alexa-488, 470/22 nm excitation and 514/30 nm emission filters; Alexa-555, 556- to 570 nm excitation and 600- to 650 nm emission filter; Alexa-633, 624- to 664 nm excitation and 692- to 732 nm emission filter) (Figs. 1b, e, 6b and Supplementary Figs. 8a, 14b). Fluorescence images (Fig. 3a) were obtained with a laser scanning confocal microscope equipped with a 100× (1.35 NA) oil immersion objective (LSM710; Zeiss), and images were collected with ZEN software. Fluorescence images (Fig. 3b and Supplementary Fig. 8b) were obtained with a laser scanning confocal microscope equipped with a HC PLAPO CS2 100× oil immersion objective (TCS SP8; Leica), and images were collected with Fiji software.

**Immunoelectron microscopy**. Hippocampal neurons grown on coverslips were transduced with lentiviruses carrying either SypHy or Stx7-SEP, and were fixed for 2 h at RT with 4% paraformaldehyde and 0.05% glutaraldehyde in 0.1M phosphate

buffer (PB) (pH7.4) at 16 DIV. These neurons were processed for subcellular localization of SypHy or Stx7-SEP with an anti-GFP antiserum by employing pre-embedding immunoelectron microscopy. Briefly, plasma membranes of fixed neurons were permeabilized with 0.1% Triton X-100 in phosphate-buffered saline (PBS) for 15 min followed by blocking with 10% fetal bovine serum in PBS for 30 min and then incubated with rabbit anti-GFP antiserum (1:1,000 dilution in PBS containing 10% fetal bovine serum) for 24 h at 4 °C. After several washes with PBS, they were incubated with 1.4 nm immunogold-conjugated anti-rabbit IgG nanoprobes (1:100 dilution in 2% normal goat serum in PBS) overnight at 4 °C and then subjected to post-fixation with 1% glutaraldehyde in PBS for 10 min. Immunogold particles were enhanced for 7 min using the HQ silver enhancement kit (Nanoprobe) according to the manufacturer's instructions. After post-fixation with 1% osmium tetroxide in PB for 40 min on ice, these samples were stained with 1% uranyl acetate solution for 35 min, dehydrated with an ethanol series and propylene oxide for 10 min each, and then embedded in Durcupan resin. After polymerization of the resin at 60 °C for 48 h, coverslips were removed and serial ultrathin sections (70 nm thick) were prepared. These sections were observed under a transmission electron microscope at 100 kV (H-7650, Hitachi Co., Japan) and serial images of individual presynaptic varicosities, which were identified by thickenings of axons with SV clusters and by postsynaptic counterparts with electron-dense thickenings of the plasma membrane (postsynaptic density: PSD) with a constant gap between the two profiles, were taken with a CCD camera (Quemessa; Olympus-SIS, Germany). In quantification of immunolabeling, one image, in which PSD was clearly seen, from serial images of individual presynaptic varicosities was used and only immunoparticles located within the presynaptic profiles were considered as having originated from SEPs expressed in SVs or early endosomes to exclude signals from SEPs expressed on the plasma membrane. The area of presynaptic varicosities, the number of immunoparticles within the individual varicosities were measured. In order to measure the nearest distance to the AZ membrane for individual immunoparticles, the AZ membrane was defined as a portion of the presynaptic plasma membrane in apposition to the PSD, and vertical lines were manually drawn from the edges of the AZ and then only immunoparticles located within the enclosed areas were used for measurements (Supplementary Fig. 9). All of the above measurements were carried out using iTEM software (Olympus-SIS).

**Protein purification**. Expression of GST-tagged Stx7-NTD or Syb2-ΔTMD was induced in *Escherichia coli* BL21 cells by adding 1 mM isopropyl-β-D-thiogalactopyranoside (IPTG) at 37 °C for 4 h for GST-Stx7-NTD, or 0.1 mM IPTG at 25 °C, overnight for GST-Syb2-ΔTMD. Recombinant proteins were purified with glutathione-Sepharose 4 Fast Flow (Amersham Biosciences) resin according to the manufacturer's instructions. Protein concentrations were determined by BCA Assay (Pierce).

**Immunoblotting**. Protein samples were separated on SDS-polyacrylamide mini-gels and blotted onto PVDF membranes (Millipore). As primary antibodies, rabbit polyclonal anti-Stx7 (Bethyl Laboratories), mouse monoclonal anti-Syb2 (Cl69.1), and mouse monoclonal anti-Tuj1 (Covance) were used. Blots were further incubated with secondary antibodies (anti-mouse IgG or anti-rabbit IgG, respectively) coupled to horseradish-peroxidase and developed using a Western Lightning Plus-ECL kit (PerkinElmer). Signals were detected using a Molecular Imager ChemiDoc (BioRad).

To estimate the copy number of Stx7 per SV, the CPG-purified SV fraction (a gift from Reinhard Jahn) with known protein concentration and vesicle number, was subjected to western blotting together with various amounts of standard proteins for Stx7 and Syb2. Band intensities in acquired images were quantified using Quantity One software (BioRad).

**Animals**. Pregnant ICR mice were purchased from SLC, Japan. All mice were given food and water *ad libitum*. Animals were kept in a local animal facility with a 12 h light/12 h dark cycle. Ambient temperature was maintained around at 21 °C with a relative humidity of 50%. All animal experiments were approved by the Institutional Animal Care and Use Committee of Doshisha University.

**Statistics**. For imaging experiments, unpaired *t*-tests were applied to compare the means of two experimental groups. All statistical tests were two-tailed, and the level of statistical significance is indicated by asterisks: $*p < 0.05$, $**p < 0.01$, $***p < 0.001$. n.s., not significant. To evaluate the significance of distances from immunoparticles to the nearest AZ membrane, a Kolmogorov−Smirnov test was performed.

**Reporting summary**. Further information on research design is available in the Nature Research Reporting Summary linked to this article.

## Data availability
All relevant data that support the findings of this study are available from the corresponding authors upon request. Detailed data for Figs. 1e–h, 2c, d, 3d–f, 4, 5, 6c−g, and 7b, c are included in Supplementary Data 1.

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

## Acknowledgements

We thank all members of the Takamori Laboratory and Dr. Tetsuya Hori for helpful discussions and Takako Maegawa at the University of Fukui for technical assistance in EM analysis. We also thank Dr. Steven D. Aird [www.sda-technical-editor.org] for editing the paper. This work was supported by grants from JSPS KAKENHI (16H04675, 19H03330), the JSPS Core-to-Core program, A. Advanced Research Networks (grant number: JPJSCCA20170008), and the Naito Foundation to S.T., grants from JSPS KAKENHI (16H06280 and 18H05120) to Y.F., and a grant from JSPS KAKENHI (18K06473) to Y.M.

## Author contributions

Y.M. and S.T. conceived the study. Y.M., K.T. and Y.F. performed experiments and analyzed the data. S.T. wrote the paper with assistance from Y.M. and Y.F., Y.M. and S.T. made the figures.

## Competing interests

The authors declare no competing interests.
