## [Peer Review File · Communications Biology]

Reviewers' Comments:

Reviewer #1:

Remarks to the Author:

Mori et al report that synaptic vesicles that are labeled by Stx7-SEP are recruited during intense high-frequency stimulation (>20Hz) but not low-frequencies (10 Hz). This was different than the observation for "regular" recycling pool vesicles labeled with syphy, which fuse at all frequencies tested.

However, syphy labeled vesicles and stx7-SEP labeled ones had important features in common – their resting pH was similar and their response to stimulation could be fully annulled by TeNT, which cleaves the SNARE VAMP2. Unlike syphy, which was seen by immuno-EM quite homogeneously throughout the vesicle cluster, stx7-SEP was observed far from the membrane. Vesicles labeled with other SEP-fused syntaxins (stx6,8,12/13,16), as well as vti1a and VAMP7 (which have been suggested to mark a different population of vesicles), had a higher resting pH, and were only partially (or not at all) affected by TeNT, implying that these SNAREs were on vesicles that indeed differ from syphy-labeled ones (i.e., not conventional synaptic vesicles). On this basis, the authors conclude that Stx7-SEP vesicles are a subgroup of the recycling pool. They show that when extracellular calcium is in the physiological range, fusion of syphy vesicles stimulated at 10 Hz is unaffected to latrunculin, whereas at 40Hz latrunculin does inhibit release. In contrast, stx7-SEP vesicles are sensitive to latrunculin at both frequencies. When the extracellular calcium concentration was increased to 8 mM, the effect of latrunculin on syphy vesicles equalized to that of stx7-SEP vesicles, i.e., latrunculin had an effect under both frequencies. Furthermore, stx7-SEP vesicles became much more responsive to 10Hz stimulation. To conclude – intense activity, mimicked by increasing extracellular calcium, made stx7-SEP vesicles available for release at lower frequencies, and revealed the latrunculin-sensitivity of syphy vesicles.

When the N-terminus segment of stx7-SEP is removed, stx7-SEP behaves like the other Stx-SEPs in this study (stx6,8,12/13,16), i.e., it does not differentiate between 10 and 40 Hz, it becomes only partially sensitive to TeNT and the pH of the vesicles is higher. This implies that in the absence of the N-terminus, stx7 is targeted to a different type of intracellular organelle. Finally, the authors showed that expressing the soluble N terminus of Stx7 can slow the kinetics of syphy-reported exocytosis, and that this effect occludes sensitivity to latrunculin. This is consistent with the N-terminus of stx7 inhibiting specifically the function of Stx7 containing vesicles. This is an elegantly performed and quite convincing study. The results and conclusions are important. Furthermore, a role for stx7 in synaptic function is somewhat surprising, considering stx7 has been mostly discussed in the literature in respect to endosome traffic in non-neuronal cells.

Nevertheless, in my opinion, several topics should be clarified further regarding the role the authors propose Stx7 plays in SV recycling in order to increase the impact of the current manuscript:

1. Do stx7-SEP vesicles indeed fuse with the plasma membrane?

Stx7-SEP fluorescence clearly changes during stimulation. It is assumed by the authors, but is not conclusively shown or stated, that the change in fluorescence arises due to fusion of stx7-SEP-carrying vesicles with the plasma membrane. However, it could be argued that these vesicles fuse with a neutral unidentified intracellular compartment. I feel that the manuscript can be strengthened by proving that plasma membrane fusion occurs, in order to convince the reader that the phenomenon that is being imaged is indeed exocytosis of synaptic vesicles. I suggest applying pH 5.5 extracellular solution after the onset of exocytosis to show that a substantial fraction of the stimulation-induced fluorescence is quenched, indicating that it is extracellular (as in Fig 2 of 10.1038/35008615). If the results are not conclusive enough because of ongoing endocytosis, the same experiment can be performed in the presence of an endocytosis inhibitor (dynasore?). The same experiment should be done for syphy, as a reference for SV exocytosis.

2. Are stx7-SEP vesicles a special subgroup of the total recycling pool labeled with syphy, or are they a separate group?

To address this question, the authors can stimulate the syphy-expressing neurons exhaustively at 10 Hz in the presence of bafilomycin, conditions which according to their experiments do not recruit stx7-laden vesicles. Fluorescence should increase to a plateau. At this point, the authors can further stimulate the neurons at 40Hz. If an additional increase in fluorescence is observed, this will indicate that there is a separate subpopulation of syphy-marked vesicles which is made

available for release by higher frequencies. The same experiment can be repeated with stx7-SEP, in which case no fluorescence is expected to be seen in the first segment, but is expected in the second. As a bonus, this experiment may reveal by a different means the proportion of stx7-controlled SVs out of the total recycling pool (in the syphy experiments).

3. If stx7 vesicles are a subgroup of the recycling pool, there are several quantitative aspects related to the size of the recycling pool that need to be reconciled.

Quantitatively, I see a few problems with the authors' suggestion that stx7 vesicles are a subgroup of the recycling pool. For example, in figure 4d, latrunculin decreased the kinetics of recycling pool SV recruitment and fusion, but not its size. If stx7 SVs are actin-dependent, and they are a subgroup of the recycling pool, I would expect a quantitative decrease in the cumulative fluorescence.

4. The structure of the title of the manuscript is confusing. The verb is "confers" which means "grants or bestows" (a property?). It is unclear to me what is conferred on the "actin-dependent rapidly replenishing synaptic vesicles". Please rephrase.

5. To authors show that expression of the N-terminus of stx7 affects neurotransmission (fig. 6). Does overexpression of stx7 itself affect neurotransmission? In this respect, do the authors have any indication whether the N-terminus of stx7 interact with stx1?

Minor points:

1. I would suggest adding VAMP2 to Fig. 7e, to clarify to the reader that stx7 appears in a subgroup of the recycling pool.

2. Can the author conjecture as to what other SNAREs interact with stx7 in the context of this paper? VAMP8? Others?

3. Can the authors explicitly state whether they believe that stx7 plays a direct role in SV fusion? Or do they suggest that it rather serves in a recycling decision point that produces a subset of SVs that differ in their properties?

4. In some manuscripts, the recycling pool is defined as containing the RRP as well (L. 37). Do the authors define the RRP and the recycling pools as separate pools?

5. L. 60: other families of proteins have been credited with playing roles in differentiating recycling and resting pools, such as the synapsins (see for example: 10.1523/JNEUROSCI.5058-11.2012)

6. In fig. 2, super-resolution would have been more convincing. At the very least, labeling active zone proteins (bassoon? Munc 13?) is advisable, to show the directionality of the presynaptic terminal. Furthermore, the EM images provided here were of insufficient quality to convince me that the labeling is associated with vesicles rather than other intracellular structures like endosomes.

7. I would point out that VAMP2 is not the only target of TeNT. Cellubrevin is also a target (10.1111/j.1600-0854.2005.00288.x), as is VAMP1, which is expressed in a subset of neurons and can substitute for VAMP2 (10.1152/jn.00340.2014). I do not think that these observations invalidate the authors' claims, though.

8. I was surprised that the authors didn't use mOrange to quantify vesicle pH, which they showed in a different manuscript to be more accurate in this respect (10.1073/pnas.1604527113).

9. The authors used the P2A peptide to co-express syphy and the N-terminus of stx7. I would suggest verifying by Western blot that self-cleavage was complete. I have observed in my lab cases where the P2A peptide did not function as expected.

10. L. 377. I would cite 10.1038/nature13846 in this context.

11. I identified a few grammar errors. I would suggest the authors scan the paper for others:

a) L. 20: ...comprehensive optical imaging for various presynaptic... "for" should be "of"

b) L. 25: ...disruption of Stx7 function by overexpressing the N-terminal domain... "the" should be "its" (the same in L. 80)

c) L. 27: ...essential for adaptation of synapses to respond high frequency... insert "to" before "high"

d) L. 45: ...RRP is believed to... insert "the" before RRP

e) L. 76: add "a" before "subpopulation"

f) L. 150: instead of "trials" use "attempts"

g) L. 501: take -> taken

h) L. 504: For each constructs -> For each construct,

Reviewer #2:

Remarks to the Author:

Though the mechanisms underlying synaptic vesicle (SV) recycling have been studied extensively, molecular details are still missing, though the involvement of distinct pools and recycling mechanisms is acknowledged. In this work, Mori and colleagues demonstrates the involvement of the SNARE protein syntaxin 7 (Stx7) in one of the mechanisms of SV recycling, a mechanism that occurs during high-frequency repetitive stimulation in hippocampal neurons and is actin-dependent. Moreover, the authors show that high-frequency stimulation can be replaced by high Ca^{2+} , implying that Stx7 might take place in the Ca^{2+} /calmodulin-dependent fast SV replenishment that was demonstrated in calyx of Held synapses. Indeed, in a similar manner to the Ca^{2+} /calmodulin-dependent SV replenishment, also the Stx7-mediated process is actin dependent. Therefore, this paper addresses an important question and identifies a key player in one of the recycling mechanisms. The paper is very well written. The rationale for the experiments and the experimental outline are clearly presented and the results are convincing.

I have a few suggestions that may strengthen the association of Stx7 with the suggested recycling pool of SVs.

1) Given the similarities with the Ca^{2+} /calmodulin-dependent fast SV replenishment, it will be interesting to see how inhibition of calmodulin affects the fluorescence responses of Stx7-SEP to high-frequency stimulation.

2) The authors assign Stx7 to a specific subset of endosomes that replenish SVs and their EM picture (Fig. 2b) shows a defined pattern of Stx7 distribution as well as colocalization with SypHy (Fig. 2a). They also present functional evidence according to which Stx7- Δ NTD-SEP is sorted to a distinct compartment than Stx7-SEP and accordingly does not function. Does Stx7- Δ NTD-SEP colocalize with SypHy? It would help to quantify colocalization of Stx7-SEP in comparison with Stx7- Δ NTD-SEP/ SypHy. Does it show a different distribution pattern?

3) The authors argue that Stx7 is targeted to a specific subset of recycling vesicles that replenish SVs only under defined conditions of stimulation, yet the results of Stx7 silencing imply that it plays an essential role in the basal biogenesis of the SVs. The authors should elaborate on this point-is it the soma localized Stx7 that is responsible for the biogenesis? At what time point was Stx7- Δ NTD-SEP introduced into the cells?

Responses to Reviewer #1

We very much appreciate positive comments and constructive suggestions on our manuscript. Accordingly, we have performed several key experiments as outlined below. The corresponding parts, as well as other changes in response to the editor and Reviewer #2, are highlighted in yellow in the revised manuscript, so that changes can be easily identified.

1. Do stx7-SEP vesicles indeed fuse with the plasma membrane?

Stx7-SEP fluorescence clearly changes during stimulation. It is assumed by the authors, but is not conclusively shown or stated, that the change in fluorescence arises due to fusion of stx7-SEP-carrying vesicles with the plasma membrane. However, it could be argued that these vesicles fuse with a neutral unidentified intracellular compartment. I feel that the manuscript can be strengthened by proving that plasma membrane fusion occurs, in order to convince the reader that the phenomenon that is being imaged is indeed exocytosis of synaptic vesicles. I suggest applying pH 5.5 extracellular solution after the onset of exocytosis to show that a substantial fraction of the stimulation-induced fluorescence is quenched, indicating that it is extracellular (as in Fig 2 of 10.1038/35008615). If the results are not conclusive enough because of ongoing endocytosis, the same experiment can be performed in the presence of an endocytosis inhibitor (dynasore?). The same experiment should be done for syphy, as a reference for SV exocytosis.

We appreciate this suggestion. Indeed, we agree that the possibility that Stx7-SEP-laden vesicles fuse with neutral intracellular organelles should be excluded. To verify this, we performed acid quenching experiments suggested by the reviewer, and obtained results clearly indicating that Stx7-SEP vesicles indeed fuse to the plasma membrane, since fluorescence was largely quenched by acid application. These results are now incorporated in Supplementary Fig. 5. We omitted the same experiments with SypHy, since we rigorously performed the same experiments to deduce dynamics of SV pH after endocytosis (Egashira et al., J Neurosci, 2015). We simply cited this paper.

P. 7

'We ruled out the possibility that stimulus-dependent increases in Stx7-SEP fluorescence resulted from vesicle fusion to neutral intracellular compartments during HFS, since application of an acidic solution (pH 5.5) right after cessation of stimulation largely quenched the fluorescence Stx7-SEP (Supplementary Fig. 5), as was observed in the case of SypHy¹⁹.'

2. Are stx7-SEP vesicles a special subgroup of the total recycling pool labeled with syphy, or are they a separate group?

To address this question, the authors can stimulate the syphy-expressing neurons exhaustively at 10 Hz in the presence of bafilomycin, conditions which according to their experiments do not recruit stx7-laden vesicles. Fluorescence should increase to a plateau. At this point, the authors can further stimulate the neurons at 40Hz. If an additional increase in fluorescence is observed, this will indicate that there is a separate subpopulation of syphy-marked vesicles which is made available for release by higher frequencies.

The same experiment can be repeated with stx7-SEP, in which case no fluorescence is expected to be seen in the first segment, but is expected in the second. As a bonus, this experiment may reveal by a different means the proportion of stx7-controlled SVs out of the total recycling pool (in the syphy experiments).

We are grateful for these valuable suggestions. We had performed SypHy experiments before submission, but we had not tested Stx7-SEP with the same paradigm. Unfortunately, the new results are somewhat confusing, and difficult to interpret at a glance. Essentially, neurons expressing either SypHy or Stx7-SEP were stimulated at low frequency (5 Hz) until the total recycling pool labeled with SypHy was depleted, and then subjected to higher stimulation frequency (40 Hz). Whereas SypHy fluorescence only slightly increase by 40-Hz stimulation, Stx7-SEP fluorescence increased dramatically by 40-Hz stimulation. A simple explanation would be that Stx7-SEP represents only very minor portion of the total SypHy-positive recycling pool. However, surprisingly, the second Stx7-SEP response was largely insensitive to TeNT-treatment, incompatible with observations regarding the absence of 5-Hz pre-stimulation shown in Fig. 1a and Supplementary Fig. 2, in which the Stx7-SEP response at 40 Hz was almost completely abolished by TeNT. Although these unexpected results do not accord with our model, we believe that these experiments suggested by this reviewer are legitimate; therefore, the results should be presented for the sake of scientific integrity. Accordingly, we present these new results in Supplementary Fig. 7, and the possible interpretation is described in the text as below.

P. 8

'The reluctant, and incomplete recruitment of Stx7-SEP vesicles during prolonged stimulation at low frequencies (5 or 10 Hz) allows us to examine whether Stx7-SEP vesicles represent a subgroup of the total recycling pool labeled with SypHy, or whether they are separate groups, although TeNT treatment clearly abolished Stx7-SEP responses within a restricted time frame (Fig. 1g). To this end, neurons expressing either SypHy or Stx7-SEP were subjected to low-frequency stimulation at 5 Hz, for 500 APs, in order to deplete the total recycling pool of SypHy, and then subsequently subjected to high-frequency stimulation at 40 Hz, 600 APs (Supplementary Fig. 7). The second stimulation at 40 Hz produced a scant increase in SypHy fluorescence, whereas the same stimulation produced a drastic increase in Stx7-SEP, indicating that Stx7-SEP vesicles represents a negligible portion of the total recycling pool. However, pretreatment of neurons with TeNT did not completely abolish the Stx7-SEP response to the second stimulation at 40 Hz, incompatible with results in the absence of prolonged pre-stimulation at 5 Hz (Fig. 1g, Supplementary Fig. 2), strongly indicating that Stx7-SEP vesicles recruited for release under this condition were not typical SVs. These unexpected observations can be explained if a substantial shift of Stx7-SEP from the SV pool to the non-SV pool occurs, perhaps mediated by fusion events via Stx7 and other endosomal SNAREs in non-SV compartments during the prolonged stimulation at 5 Hz. The unsolved question of whether Stx7-SEP vesicles comprise part of the total recycling pool will be addressed by other approaches, as described below (see Figs. 6 and 7).'

3. If stx7 vesicles are a subgroup of the recycling pool, there are several quantitative aspects related to the size of the recycling pool that need to be reconciled.

Quantitatively, I see a few problems with the authors' suggestion that stx7 vesicles are a subgroup of the recycling pool. For example, in figure 4d, latrunculin decreased the kinetics of recycling pool SV recruitment and fusion, but not its size. If stx7 SVs are actin-dependent, and they are a subgroup of the recycling pool, I would expect a quantitative decrease in the cumulative fluorescence.

As shown in Figure 4d, latrunculin decreased the kinetics, but not the magnitude of SV recruitment, indicating that inhibition of actin polymerization simply attenuates the speed of rapidly replenishing SVs, but these SV components can still undergo replenishment at slower rates (which might be similar to those of slowly replenishing SVs proposed by Sakaba and Neher (Neuron, 2000). This interpretation is compatible with observations in calyx of Held synapses, in which latrunculin only slows kinetics of SV replenishment after

RRP depletion. This is evident from the fact that EPSCs recovered completely even after latrunculin treatment.

We also realize that some Stx7-positive vesicles that are not genuine SVs also exist at the terminals (since they undergo exocytosis in a VAMP-independent manner during prolonged stimulation). This is now stated clearly as above.

4. The structure of the title of the manuscript is confusing. The verb is “confers” which means “grants or bestows” (a property?). It is unclear to me what is conferred on the “actin-dependent rapidly replenishing synaptic vesicles”. Please rephrase.

Good suggestion! Our new experiments during the revision revealed additional information regarding signaling downstream of Ca^{2+} . Accordingly, we simplified the title to ‘The endosomal Q-SNARE, Syntaxin 7, defines a rapidly replenishing synaptic vesicle recycling pool in hippocampal neurons’.

5. To authors show that expression of the N-terminus of stx7 affects neurotransmission (fig. 6). Does overexpression of stx7 itself affect neurotransmission? In this respect, do the authors have any indication whether the N-terminus of stx7 interact with stx1?

There are, at least, two reasons that make us believe that Stx7 does not have a profound influence on neuronal SNARE complex function (e.g. through inactivation of Stx1). First, overexpression of Stx7-SEP did not produce drastic changes in Syp-mOr responses (Supplementary Fig. 6), which would have been altered if Stx7 itself affects neurotransmission. Second, the SypHy response at 10-Hz stimulation was unaltered upon expression of the N-terminus of Stx7 (Fig. 7b), excluding the possibility that Stx7-N-term directly regulates any of the neuronal SNAREs.

Minor points:

1. I would suggest adding VAMP2 to Fig. 7e, to clarify to the reader that stx7 appears in a subgroup of the recycling pool.

According to the suggestion, we have added multiple copies of VAMP2 in green to Fig. 7d.

2. Can the author conjecture as to what other SNAREs interact with stx7 in the context of this paper? VAMP8? Others?

As the editor also pointed out, we have screened additional putative SNARE pairs of Stx7 during our revision. We had tested Stx8 and VAMP7 in the original manuscript, because they are known to be enriched in the SV fraction. Here, we extended the analysis to VAMP8 and vti1b, and found that they do not behave like Stx7-SEP. Thus, we can not conjecture which SNAREs interact with Stx7 at synapses, which indeed highlights a peculiarity of Stx7 among cognate endosomal SNAREs. Nevertheless, the new results are incorporated in Supplementary Fig. 4 and described in the main text as follows.

P.6

*We also examined additional putative SNARE pairs of Stx7 suggested in non-neuronal cells, including VAMP8 and vti1b^{26,27}. When VAMP8-SEP and vti1b-SEP were lentivirally transduced into neurons, they rarely co-localized with Syb2/VAMP2-positive puncta, and only minor portions of

punctate structures labeled with these SNAREs responded to repetitive stimulation at 40-Hz, 200 APs (~11% of VAMP8-SEP-positive puncta and ~0.7% of vti1b-SEP-positive puncta showed fluorescence increases beyond 5% of total SEP fluorescence revealed by NH₄Cl application, whereas ~77% and ~94% of Stx7-SEP-positive puncta and SypHy-positive puncta, respectively, showed responses beyond 5% of their total fluorescence) (Supplementary Fig. 4).'

3. Can the authors explicitly state whether they believe that *stx7* plays a direct role in SV fusion? Or do they suggest that it rather serves in a recycling decision point that produces a subset of SVs that differ in their properties?

Since effects of disruption of Stx7 function (by overexpressing the Stx7 N-terminus) cause changes in kinetics of SypHy only during relatively high-frequency stimulation, we believe that Stx7 does not play a direct role in SV fusion events. Rather, as expected for its SNARE function in intracellular membrane fusion, we believe that membrane fusion involving Stx7 is responsible for producing a subset of SVs that are recruited for release rapidly upon HFS. Nonetheless, pHluorin imaging is not sensitive enough to resolve a single-SV fusion events; therefore, we cannot explicitly exclude the possibility that Stx7 is directly involved in SV exocytosis, such as tethering, docking, priming, or fusion.

To clarify our intent, we revised the last sentence of the abstract.

Abstract

'Thus, our data indicate that endosomal membrane fusion involving Stx7 forms rapidly replenishing vesicles essential for synaptic responses to high-frequency repetitive stimulation.'

4. In some manuscripts, the recycling pool is defined as containing the RRP as well (L. 37). Do the authors define the RRP and the recycling pools as separate pools?

Yes. Although the border between RRP and the recycling pool with pHluorin-imaging is not clear, RRP is believed to be a separate from the general recycling pool. Perhaps, the RRP is a part of 'total recycling pool' that includes RRP and recycling pool (Alabi and Tsien, *Cold Spring Harb. Perspect. Biol.*, 2012; Rizzoli and Betz, *Nat. Rev. Neurosci.*, 2005).

5. L. 60: other families of proteins have been credited with playing roles in differentiating recycling and resting pools, such as the synapsins (see for example: 10.1523/JNEUROSCI.5058-11.2012)

We added the text below to the Introduction.

'Indeed, in addition to synapsins, which are reportedly associated with RP⁸, endosomal SNAREs are also present in distinct vesicle pools, namely in RP or spontaneously releasing vesicles (VAMP7 or vti1a), both of which are somewhat refractory to evoked release^{9,10}.'

6. In fig. 2, super-resolution would have been more convincing. At the very least, labeling active zone proteins (*bassoon?* *Munc 13?*) is advisable, to show the directionality of the presynaptic terminal.

In order to gain further insights into endogenous Stx7 localization at presynaptic terminals, we performed triple immunostaining to localize Syp and Stx7 relative to an active zone marker, *bassoon*. Even with conventional confocal microscopy, we observed that although Syp immunoreactivities were localized in close proximity to *bassoon* signals, in which *bassoon*-positive signals often located at the center of Syp puncta, Stx7

immunoreactivity did not closely contact bassoon signals. The existence of Stx7 outside SV clusters is consistent with the observation that Stx7 is also present in vesicles whose exocytosis is not sensitive to TeNT treatment upon prolonged stimulation.

Furthermore, the EM images provided here were of insufficient quality to convince me that the labeling is associated with vesicles rather than other intracellular structures like endosomes.

One of the intrinsic problems of immuno-electron microscopy for intracellular proteins is the difficulty of preserving membrane structures upon detergent treatment for antibody penetration. We think that the quality of our EM images is comparable to previous results for other SV proteins published from other laboratories (Hua, et al., *Neuron*, 2011; Ramirez, et al., *Neuron*, 2012). Unfortunately, none of the EM images, including ours, allow us to distinguish SVs from endosomes of similar sizes. Nevertheless, we should mention that, in response to another reviewer, we repeated EM analysis and quantified the densities and distances of immune-gold labeling for SypHy and Stx7-SEP from the active zone (Figure 3c-f and Supplementary Figure 9). Accordingly, we added descriptions of the results as follows:

P. 8-9

Stx7 localizes to a subset of SVs with low abundance at the presynaptic terminals

In order to gain further evidence for localization of Stx7 in a subpopulation of SVs, we adopted morphological and biochemical approaches. First, we asked whether endogenous Stx7 localizes to a specific part of the SV cluster. To this end, we co-stained cultured hippocampal neurons with antibodies against Stx7 and Syp, the latter of which should illuminate entire SV clusters. Although Stx7 immunoreactivity was apparent in cell somas and dendrites, it was also observed along axons (Supplementary Fig. 8a). Images at higher magnification revealed that Stx7 fluorescence partially overlapped with Syp signals (Fig. 3a). We then performed triple staining, including an active zone (AZ) marker bassoon (BSN). Whereas BSN signals often appeared as small, compact structures located at the center of Syp-positive puncta, Stx7 immunoreactivity only partially overlapped with BSN signals and located surrounding BSN signals (Fig. 3b, Supplementary Fig. 8b), indicating that intrinsic Stx7 localizes at the distal side of the SV cluster from the AZs. To substantiate these observations, we next performed immunoelectron microscopy. Since our initial attempts to detect endogenous Stx7 with the same antibody did not produce reliable signals, we expressed Stx7-SEP as performed for SEP imaging, and proceeded to immunostaining using anti-GFP antibody. In comparison, we adopted cultured cells transfected with SypHy, which could be detected with the same antibody. Consistent with immunofluorescence data, immunoparticles for SypHy were widely spread all over SV clusters in presynaptic structures, whereas Stx7-SEP immunoparticles were sparsely distributed within SV clusters (Fig. 3c, Supplementary Fig. 9). Quantification of densities of immunoparticles revealed that SypHy was expressed at significantly higher level than Stx7-SEP (approximately 6-fold; 294 ± 33 particles / μm^2 for SypHy vs. 47 ± 5 particles / μm^2 for Stx7-SEP), while the numbers of both immunoparticles showed a positive correlation with areas of synaptic varicosity (Fig. 3d, e, Supplementary Fig. 9). Notably, distances of Stx7-SEP immunoparticles to the nearest AZ membranes (defined by electron-dense postsynaptic density (PSD) structures) were significantly longer than those of SypHy, which was evident from a right shift of a cumulative plot for Stx7-SEP compared to that for SypHy (Fig. 3f). These observations were fully compatible with our immunofluorescence data, as well as with a previous observation under STED microscopy⁷.

7. I would point out that VAMP2 is not the only target of TeNT. Cellubrevin is also a target (10.1111/j.1600-0854.2005.00288.x), as is VAMP1, which is expressed in a subset of neurons and can substitute for VAMP2 (10.1152/jn.00340.2014). I do not think that these observations invalidate

the authors claims, though.

We appreciate this comment. We have added references (Proux-Gillardeaux et al, *Traffic*, 2005; Zimmermann et al, *J. Neurophys.* 2014), and modified the manuscript at line 127.

8. I was surprised that the authors didn't use mOrange to quantify vesicle pH, which they showed in a different manuscript to be more accurate in this respect (10.1073/pnas.1604527113).

If our main aim had been to measure accurate luminal pHs of the respective SNARE-laden vesicles, we agree that mOrange2 should have been used. However, we sought to measure average luminal pH of the respective SNARE-laden vesicles exclusively at 'active' synapses. As stated in the text, the majority of endosomal SNAREs expressed via lentiviral transduction localized not only in synaptic boutons, but also in the soma and in some cases, in the dendrites. To identify active synapses by monitoring responses to electrical stimulation, pHluorin is far superior to mOrange2, because it gives a much higher S/N ratio than mOrange2. Since exocytotic responses of some endosomal SNARE-SEP turned out to be much smaller than those of SEPs fused to authentic SV proteins (see Supplementary Fig. 3), we chose to use SEP constructs to measure vesicular pH, and at the same time, to identify active synapses with higher sensitivity.

9. The authors used the P2A peptide to co-express syphy and the N-terminus of stx7. I would suggest verifying by Western blot that self-cleavage was complete. I have observed in my lab cases where the P2A peptide did not function as expected.

Indeed, we have tried to detect SypHy and the N-terminus of Stx7 by western blot using either anti-Synaptophysin antibody (CI 7.2; a mouse monoclonal antibody frequently used for synaptophysin) and anti-Stx7 antibody in order to validate P2A self-cleaving activity. Needless to say, if P2A does not work properly, we should see a band which corresponds to a fusion protein of SypHy and N-terminus of Stx7. Unfortunately, we could not detect even the expected band for SypHy and the N-terminus of Stx7. This is probably due to the fact that the average copy number of SypHy / SV was very low (only 2-3 copies of SypHy / SV, while 30 Synaptophysin copies / SV), and transduction efficiency with our method approximated ~50% (Egashira *et al.*, *PNAS*, 2016). It is, therefore, estimated that expression of SypHy represents only ~5% of endogenous Synaptophysin, which may well have eluded detection. Thus, due to the lack of sensitivity of our validation methods, we cannot rule out the possibility that the effect of Stx7-N on SypHy recycling is due to Stx7-N fragments that remained attached to the C-terminus of SypHy, owing to inefficient self-cleavage by the P2A peptide.

Although they do not completely rule out incomplete self-cleavage by the P2A peptide in our system, our experiments for a separate study suggest that the P2A peptide functions as expected. Experiments utilizing P2A self-cleavage activity were performed in order to show that a kinase named Aak1 (named after adaptor-associated protein kinase 1) is localized at presynaptic boutons in cultured hippocampal neurons. To this end, we constructed SypHy-P2A-TagRFP-Aak1 (TagRFP-fused Aak1 was placed after the P2A peptide sequence) and SypHy-P2A-TagRFP. While TagRFP-Aak1 was perfectly co-localized with SypHy, TagRFP was diffusely expressed along axons as well as cell bodies, indicating that (at least) a certain amount of TagRFP was cleaved from SypHy and separately expressed. As a reference, the figure published in *PNAS* as Fig. 5C (Taoufiq et al, *PNAS*, 2020;

www.pnas.org/cgi/doi/10.1073/pnas.2011870117) is shown below. Since this paper was published during our revision, we cited it in the Methods section.

Figure legend

(upper panels) Cultured neurons were transfected with a lentiviral vector carrying SypHy-P2A-TagRFP. Line scans at right show that SypHy was restricted to the presynaptic boutons as expected, whereas TagRFP was evenly distributed in the cytoplasm.

(Lower panels) Cultured neurons were transfected with a lentiviral vector carrying SypHy-P2A-TagRFP-Aak1. Line scans at right show that TagRFP-Aak1 preferentially localized to the presynaptic terminals where SypHy was present.

10. L. 377. I would cite 10.1038/nature13846 in this context.

We have cited the corresponding reference indicated by the reviewer. Thanks.

11. I identified a few grammar errors. I would suggest the authors scan the paper for others:

We apologize for these errors. The revised version was carefully scanned by ourselves, as well as by a professional editor, as indicated in the Acknowledgments.

Responses to Reviewer #2:

We appreciate the positive response of the reviewer and the number of valuable suggestions to improve the manuscript. During revision, we tried to address all points raised by this reviewer. In addition, we have revised the whole manuscript according to inquiries from the editor and reviewer #1, all of which are also highlighted in yellow.

1) Given the similarities with the Ca²⁺/calmodulin-dependent fast SV replenishment, it will be interesting to see how inhibition of calmodulin affects the fluorescence responses of Stx7-SEP to high-frequency stimulation.

This is very important point and we have devoted substantial effort to test how calmodulin inhibition affects the response of Stx7-SEP, as well as SypHy. First, when we incubated our cultures with the calmodulin inhibitor, calmidazolium (20 μ M for 30 min before imaging experiments as described previously by Ed Chapman's group (Liu, et al, *eLife*, 2014), neurons were gradually dying; therefore, we hardly obtained reliable exocytic responses of SEP-reporters. As an alternative approach, we sought to inhibit calmodulin activity by overexpressing calmodulin-inhibitory peptide (CIP) together with SEP-reporters (e.g. Sakaba and Neher, *Neuron*, 2000), which appeared to work better. We observed that responses of SypHy during 10-Hz stimulation were not affected by CIP, but those during 20-Hz stimulation were significantly slowed, while the total releasable pool at both stimulation frequencies was unchanged. This observation is compatible with the notion that Ca²⁺/calmodulin-dependent fast SV replenishment is preferentially triggered at intensive stimulation. Interestingly, the response of Stx7-SEP upon 20-Hz stimulation was similarly slowed by CIP, indicating that Stx7-SEP-positive SVs represent a subset of SVs that are recruited for release in a Ca²⁺/calmodulin-dependent manner. We have included these new data in Fig.4, and modified the text accordingly. Based on this change, we also modified the Abstract and Introduction, which are also highlighted in yellow.

P. 10

'To further elucidate signaling pathways downstream of Ca²⁺, we focused on calmodulin, since it is a Ca²⁺-sensor protein that mediates fast SV replenishment after RRP depletion at the calyx of Held synapse and at hippocampal synapses^{31,32}. To this end, we co-expressed the SEP reporters and a calmodulin inhibitory peptide (CIP), and monitored fluorescence changes in the presence of Baf for prolonged periods (either at 10 Hz for 60 sec or at 20 Hz for 30 sec). Consistent with previous reports, CIP expression slowed SV recruitment, measured with SypHy, during 20-Hz stimulation (Fig. 4b), whereas no changes were observed during 10-Hz stimulation (Fig. 4b), indicating that activation of the Ca²⁺/calmodulin pathway promotes rapid SV replenishment, preferentially during intensive stimulation. No changes in total pool sizes were observed at either stimulation frequency (Fig. 4b). Notably, recruitment of Stx7-SEP vesicles for exocytosis was also significantly slowed by CIP at 20-Hz stimulation (Fig. 4c). These results indicate that a calmodulin-dependent SV replenishment process preferentially operates at HSF (>20 Hz) in cultured hippocampal neurons, and that Stx7-SEP-laden vesicles may well constitute (at least a part of) such an SV pool.'

2) The authors assign Stx7 to a specific subset of endosomes that replenish SVs and their EM picture (Fig. 2b) shows a defined pattern of Stx7 distribution as well as colocalization with SypHy (Fig. 2a). They also present functional evidence according to which Stx7-ANTD-SEP is sorted to a distinct compartment than Stx7-SEP and accordingly does not function. Does Stx7-ANTD-SEP colocalize with SypHy? It would help to quantify colocalization of Stx7-SEP in comparison with Stx7-ANTD-SEP/ SypHy. Does it show a different distribution pattern?

During revision, we extended the EM analysis and analyzed images in a more quantitative manner (as was also requested by the editor and reviewer #1). We confirmed that SypHy is widely distributed in presynaptic SV clusters, whereas Stx7-SEP is distributed more sparsely than SypHy. Furthermore, Stx7-SEP immunoparticles locate further from the active zone membrane compared to SypHy immunoparticles. Together with new immunofluorescence data, including active zone protein, bassoon, we show that Stx7 is present in a subset of SVs at presynaptic terminals.

Although we have included Stx7- Δ NTD in this analysis, we could not obtain satisfactory results because of high labeling at the cell surface (consistent with SEP data in Supplementary Fig. 12) and poor labeling in SV clusters. We, therefore, decided to present EM data only for SypHy and Stx7-SEP at this point to show difference in their expression at presynaptic terminals (Fig. 3c-f, and Supplementary Fig. 9), and described as follows:

P. 9

'To substantiate these observations, we next performed immunoelectron microscopy. Since our initial attempts to detect endogenous Stx7 with the same antibody did not produce reliable signals, we expressed Stx7-SEP as performed for SEP imaging, and proceeded to immunostaining using anti-GFP antibody. In comparison, we adopted cultured cells transfected with SypHy, which could be detected with the same antibody. Consistent with immunofluorescence data, immunoparticles for SypHy were widely spread all over SV clusters in presynaptic structures, whereas Stx7-SEP immunoparticles were sparsely distributed within SV clusters (Fig. 3c, Supplementary Fig. 9). Quantification of densities of immunoparticles revealed that SypHy was expressed at significantly higher level than Stx7-SEP (approximately 6-fold; 294 ± 33 particles / μm^2 for SypHy vs. 47 ± 5 particles / μm^2 for Stx7-SEP), while the numbers of both immunoparticles showed a positive correlation with areas of synaptic varicosity (Fig. 3d, e, Supplementary Fig. 9). Notably, distances of Stx7-SEP immunoparticles to the nearest AZ membranes (defined by electron-dense postsynaptic density (PSD) structures) were significantly longer than those of SypHy, which was evident from a right shift of a cumulative plot for Stx7-SEP compared to that for SypHy (Fig. 3f). These observations were fully compatible with our immunofluorescence data, as well as with a previous observation under STED microscopy⁷.

3) The authors argue that Stx7 is targeted to a specific subset of recycling vesicles that replenish SVs only under defined conditions of stimulation, yet the results of Stx7 silencing imply that it plays an essential role in the basal biogenesis of the SVs. The authors should elaborate on this point-is it the soma localized Stx7 that is responsible for the biogenesis? At what time point was Stx7- Δ NTD-SEP introduced into the cells?

We observed presynaptic defects in Stx7-KD when shRNA was introduced at DIV 0. Furthermore, our preliminary results indicate that PSD-95 immunoreactivity decreases upon Stx7-KD when shRNA was introduced at DIV 0 (see images below). On the other hand, the reduction of Stx7 expression by shRNA could not be achieved when shRNA was introduced at DIV 7; therefore, we did not perform pHluorin imaging. Since we noticed that inhibition of Stx7 at early stages of culture caused global defects in SV vesicle recycling, irrespective of stimulation frequencies, we decided to introduce Stx7-NTD at DIV7 to avoid potential effects on synapse development, maturation or SV biogenesis. Since the possible role of Stx7 in synaptogenesis is not the main issue of this paper, we do not want to present these results here.

We have to admit that we cannot conclude at this point whether soma-localized Stx7 is responsible for the observed phenotypes, but it is likely because the effect was evident

when KD was initiated at DIV 0 compared to DIV 7 (note that synaptic connections emerge around DIV7).

REVIEWERS' COMMENTS:

Reviewer #1 (Remarks to the Author):

In my opinion, the authors have fully answered all my previous questions and comments. I see no additional issues that need attention. Overall, this is a very interesting paper and worthy of attention by our peers.

Following are a few minor comments that the authors may want to address, if at all.

I agree with the authors that their finding that Tetanus toxin does not fully inhibit Stx7-SEP signals in supplementary figure 7 is puzzling, but since they acknowledge this in the manuscript text, I accept that the readers have the option of forming their own informed opinion.

I would have preferred to see some quantification of the immunofluorescence results presented in figure 3, but since EM results are quantified, this is not a must.

I do point out that differences in the quantity of Stx7-SEP immunogold labeling could be attributed also to the expression level, and not just, as the authors explain, to the number of labeled vesicles. An analysis of expression levels for this experiment would have been helpful. However, the difference in the distribution of labeling does lend credibility to the authors' interpretation.

Reviewer #2 (Remarks to the Author):

The authors have addressed my comments to my satisfaction. I have no further comments.